# Distinct signals and immune cells drive liver pathology and glomerulonephritis in ABIN1[D485N] mice

Sambit Kumar Nanda[1,]*, Tsvetana Petrova[1,2,]*, Francesco Marchesi[3], Marek Gierlinski[4], Momchil Razsolkov[2], Katherine L Lee[5], Stephen W Wright[5], Vikram R Rao[6], Philip Cohen[1], J Simon C Arthur[2]

We report that TLR7, IL-6, and the adaptive immune system are essential for autoimmunity and glomerulonephritis but not for liver pathology in mice expressing the ubiquitin-binding–defective ABIN1[D485N] mutant. The blood and organs of ABIN1[D485N] mice have exceptionally high numbers of patrolling monocytes (pMo), which develop independently of IL-6 and the adaptive immune system. They are detectable in the blood months before autoimmunity and organ pathology are seen and may have diagnostic potential. The splenic pMo, inflammatory monocytes (iMo), and neutrophils of ABIN1[D485N] mice expressed high levels of mRNAs encoding proteins released during NETosis, which together with the high numbers of monocyte-derived dendritic cells (MoDCs) may drive the liver pathology in ABIN1[D485N] mice, and contribute to the pathology of other organs. The splenic iMo of ABIN1[D485N] mice displayed high expression of mRNAs encoding proteins controlling cell division and were actively dividing; this may underlie the increased pMo and MoDC numbers, which are derived from iMo. An orally active IRAK4 inhibitor suppressed all facets of the disease phenotype and prevented the increase in pMo numbers.

## Introduction

Systemic lupus erythematosus (SLE, lupus) is a complex disease in which the body's immune system attacks its own organs, resulting in severe inflammation and damage of these tissues. Up to 70% of lupus patients develop nephritis, which is caused by immunoglobulins and complement components becoming deposited in the glomerulus of the kidney. For this reason, studies aimed at gaining a molecular understanding of the causes of lupus have mainly focused on the pathways leading to glomerulonephritis. However, lupus affects many other organs. For example, the liver is an important target of SLE (Bessone et al, 2014), whereas 50% of lupus patients experience lung

problems, most frequently pleuritis and pneumonitis. Antinuclear antibodies (ANAs) and double-stranded DNA (dsDNA) antibodies have been detected in the pleural fluid (Porcel et al, 2007; Toworakul et al, 2011), but whether they contribute to the lung pathology seen in lupus or are just a consequence of the disease is unclear.

Genome-wide association studies have identified polymorphisms in a number of human genes that predispose to SLE. These include polymorphisms in *TNIP1*, the gene encoding A20-binding inhibitor of NF-κB1 (ABIN1), which has been reported to predispose to SLE in many human populations (Gateva et al, 2009; Han et al, 2009; Nair et al, 2009; Adrianto et al, 2012; Gregersen et al, 2012; Shi et al, 2014). A key role of ABIN1 in immune cells is to restrict activation of the protein kinases termed TAK1 (TGFβ-activated protein kinase 1) and IKKβ (IκB kinase β), preventing the overproduction of inflammatory mediators by myeloid and B cells. ABIN1 exerts this effect by interacting with the ubiquitin chains that activate TAK1 and IKKβ (reviewed (Cohen and Strickson, 2017)), as revealed by genetic studies with knock-in mice in which ABIN1 was mutated to a ubiquitin-binding–defective mutant. Stimulation of the dendritic cells or B cells from these ABIN1[D485N] mice with ligands that activate TLRs or nucleotide-binding oligomerization domain–containing receptor 2 induced the hyperactivation of TAK1 and IKKβ, causing a several-fold increase in the secretion of TNF, IL-6, and IL-12 (Nanda et al, 2011).

The ABIN1[D485N] mice are born at Mendelian frequencies and develop normally, but at 3 mo of age, develop enlarged spleens and lymph nodes with greatly increased numbers of T-follicular helper (Tfh) cells and germinal centre B (GCB) cells. At 4 mo, high levels of many immunoglobulins (Igs) appear in the serum, including ANAs and antibodies to dsDNA, followed by severe inflammation of the kidney, liver, and lungs after 5–6 mo (Nanda et al, 2011). In contrast, most ABIN1 KO mice die during embryonic development; however, the few that survive to adulthood develop a phenotype similar to but even more aggressive than ABIN1[D485N] mice (Zhou et al, 2011).

The autoimmunity and organ inflammation displayed by ABIN1 [D485N] mice is abolished by crossing to mice that do not express myeloid differentiation primary response 88 (MyD88) (Nanda et al,

[1]Medical Research Council Protein Phosphorylation and Ubiquitylation Unit, School of Life Sciences, University of Dundee, Dundee, UK  [2]Division of Cell Signaling and Immunology, School of Life Sciences, University of Dundee, Dundee, UK  [3]School of Veterinary Medicine, College of Medical Veterinary and Life Sciences, University of Glasgow, Glasgow, UK  [4]Division of Computational Biology, School of Life Sciences, University of Dundee, Dundee, Scotland, UK  [5]Worldwide Medicinal Chemistry, Pfizer Inc, New York, NY, USA  [6]Inflammation and Immunology Research Unit, Pfizer Research, Cambridge, MA, USA

Correspondence: p.cohen@dundee.ac.uk; j.s.c.arthur@dundee.ac.uk
*Sambit Kumar Nanda and Tsvetana Petrova contributed equally to this work

2011), an adaptor protein required for signaling by interleukin 1 (IL-1) family members and ligands that activate TLRs. MyD88 forms oligomeric complexes with members of the IL-1 receptor-associated kinases (IRAKs), termed the Myddosome, and lupus in ABIN1[D485N] mice is also prevented by crossing to mice in which IRAK4 or IRAK1 are replaced by kinase-inactive mutants. These studies demonstrate that components of the Myddosome are key drivers of autoimmunity in this model. In contrast, crossing to mice in which IRAK2 is replaced by a functionally inactive mutant has no effect on the progression of the disease (Pauls et al, 2013; Nanda et al, 2016).

Many human patients with lupus display high levels of IFN-stimulated genes in their peripheral blood mononuclear cells. Flt3-derived plasmacytoid dendritic cells from ABIN1[D485N] mice overproduce IFNα and IFNβ upon stimulation with ligands that activate TLR7 or TLR9, but autoimmunity is unaffected by crossing to mice that lack the type 1 IFN-associated receptor 1 subunit (IFNAR1), which is essential for type I IFN signaling. Moreover, the kidney pathology is only improved modestly in ABIN1[D485N] × IFNAR1 KO mice. The overproduction of type 1 IFNs, therefore, seems to be a consequence and not a cause of the phenotype of ABIN1[D485N] mice (Nanda et al, 2016).

Because polymorphisms in *TNIP1* predispose to human lupus and ABIN1[D485N] mice develop spontaneously a disease that closely resembles some types of human SLE (Caster et al, 2013), we have continued to investigate the molecular mechanisms driving lupus in this model. Here, we demonstrate that the MyD88-IRAK4-IRAK1 signaling axis drives both the autoimmune and autoinflammatory aspects of the lupus phenotype, as well as the increased numbers of patrolling and inflammatory monocytes and the striking changes to their gene expression profiles seen in this model.

## Results

### Autoantibody production and glomerulonephritis requires IL-6 in ABIN1[D485N] mice, but liver pathology and lung inflammation do not

IL-6 is known to stimulate the generation of splenic GCB cells (Kopf et al, 1998), which are required for isotype switching somatic hypermutation, leading to the production of high-affinity antibodies such as ANAs and anti-dsDNA autoantibodies. Both dendritic cells and B cells from ABIN1[D485N] mice show enhanced IL-6 production relative to cells from wild-type (WT) mice after stimulation with TLR-activating ligands (Nanda et al, 2011). To investigate the contribution of IL-6 to the lupus phenotype, we crossed ABIN1[D485N] mice to IL-6 KO mice and found that splenomegaly was reduced (Fig 1A) and the formation of GCB cells abolished (Figs 1B and S1A). Consistent with these observations, the levels of dsDNA antibodies, as well as the total IgM, IgG, and IgE, in the serum were reduced in ABIN1[D485N] × IL-6 KO mice relative to the ABIN1[D485N] mice (Figs 1C–E), and glomerulonephritis was strongly suppressed (Figs 1F, and S1B). However, neither the liver pathology (Figs 1G and S1C) nor lung inflammation (Figs 1H and S1D) were affected. Taken together, these experiments suggest that the overproduction of IL-6 in ABIN1[D485N] mice contributes to germinal centre formation, antibody production, and glomerulonephritis, but is not required for the liver pathology or lung inflammation seen in this model.

### The adaptive immune system is required for the development of glomerulonephritis, but not for the liver pathology of ABIN1 [D485N] mice

The results presented in Fig 1 suggested that the formation of GCB cells and increased production of autoantibodies might be the triggers for glomerulonephritis. To further investigate this possibility, we crossed ABIN1[D485N] mice to RAG2 KO mice (Shinkai et al, 1992), which lack mature B and T cells and, therefore, cannot develop GCB cells or autoantibodies. We found that glomerulonephritis measured at 6 mo of age was completely suppressed in ABIN1[D485N] × RAG2 KO mice (Figs 2A, and S2A and B). Lung inflammation was also greatly reduced, but not abolished, by crossing to RAG2 KO mice (S2C and S2C), indicating an important contribution of the adaptive immune system to the lung pathology. In contrast, the liver pathology was unaffected (S2D and S2D), indicating that it occurs independently of adaptive immunity.

The observations in ABIN1[D485N] × RAG2 KO mice suggested that the liver pathology was being driven by innate immune and/or non-immune cells. To distinguish between these possibilities, we generated bone marrow chimaeric mice by injecting equal proportions of wild-type (WT) and ABIN1[D485N] bone marrow cells into irradiated WT mice (Fig 2D). The mixed chimaeric mice developed liver pathology and splenomegaly 4 mo after the reconstitution (Figs 2E and F and S2E), which, together with the data from the cross to RAG2 KO mice, indicated an important role for haematopoietic non-T, non-B cell(s) in these processes.

As bone marrow transfer generates B and T cells, the chimaeric mice also developed aspects of the disease that are dependent on the adaptive immune system such as GCB cell formation (Figs 2G and S2F), increased ANA and anti-dsDNA antibodies (Figs 2H and I) and glomerulonephritis (Figs 2J and S2G).

### TLR7 is required for autoimmunity and glomerulonephritis in ABIN1[D485N] mice, but liver pathology and lung inflammation are less dependent on TLR7

BXSB/Yaa mice in which the TLR7 gene is duplicated develop an autoimmune disease similar to ABIN1[D485N] mice (Pisitkun et al, 2006; Santiago-Raber et al, 2009b) and the spontaneous development of lupus in several other lupus-prone mouse lines is attenuated by crossing to TLR7 KO mice (Christensen et al, 2006; Demaria et al, 2010; Santiago-Raber et al, 2010). We found that crossing the ABIN1 [D485N] mice to TLR7 KO mice prevented splenomegaly (Fig 3A), as well as the increase in splenic T$_{fh}$ cells (Figs 3B and S3A) and GCB cells (Fig 3C and S3B). Consequently, ABIN1[D485N] × TLR7 KO mice did not display increased levels of dsDNA antibodies (Fig 3D) or Igs (Figs 3E and F), and glomerulonephritis was also strongly suppressed (Figs 3G and S3C). In contrast, the liver pathology was reduced, but not abolished (Figs 3H and S3D), and the decrease in lung inflammation was not statistically significant (Figs 3I and S3E). Taken together, the results presented in Figs 1–3 indicate that autoimmunity and autoinflammation in ABIN1[D485N] mice arise from different inputs and outputs that feed into and out of the central core MyD88-IRAK4-IRAK1 signaling axis.

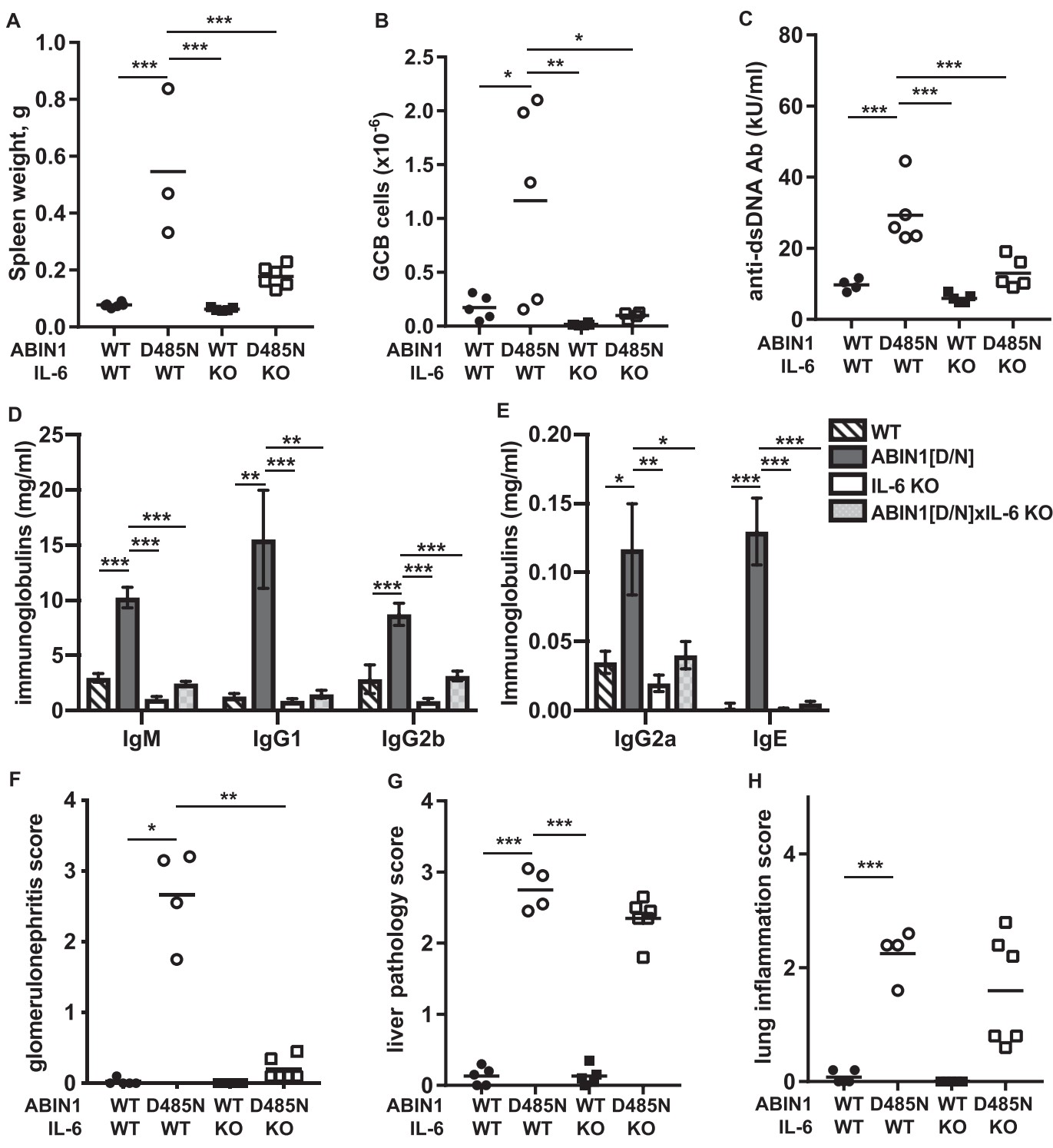

**Figure 1. Autoimmunity in ABIN1[D485N] mice, but not liver or lung inflammation, is prevented by crossing to IL-6 KO mice.**
**(A)** Spleen weights (left hand panel) of 26-wk-old WT (n = 6), ABIN1[D485N] (n = 4), IL-6 KO (n = 5), and ABIN1[D485N] × IL-6 KO (n = 6) mice. **(B)** GCB cell numbers in spleens of 17-wk-old WT (n = 5), ABIN1[D485N] (n = 5), IL-6 KO (n = 8), and ABIN1[D485N] × IL-6 KO (n = 6) mice. **(C)** Anti-dsDNA antibodies in the serum of 26-wk-old mice. **(D, E)** Same as (C), except that IgM, IgG1, and IgG2b, IgG2a and IgE concentrations were measured. **(F, G, H)** Same as (C), except that kidney (F), liver (G), and lung (H) pathology scores from WT, ABIN1[D485N], IL-6 KO, and ABIN1[D485N] × IL-6 KO mice (n = 4–6) were determined. **(A, B, C, F, G, H)** Each symbol shows the data from a single mouse. Statistical significance between the genotypes was calculated using one-way ANOVA and the Tukeys post-hoc test (A, B, C, D, G) or the Kruskal-Wallis and the Mann-Whitney tests (E, F, H);* denotes $P < 0.05$, ** $P < 0.01$ and *** denotes $P < 0.001$.
Source data are available for this figure.

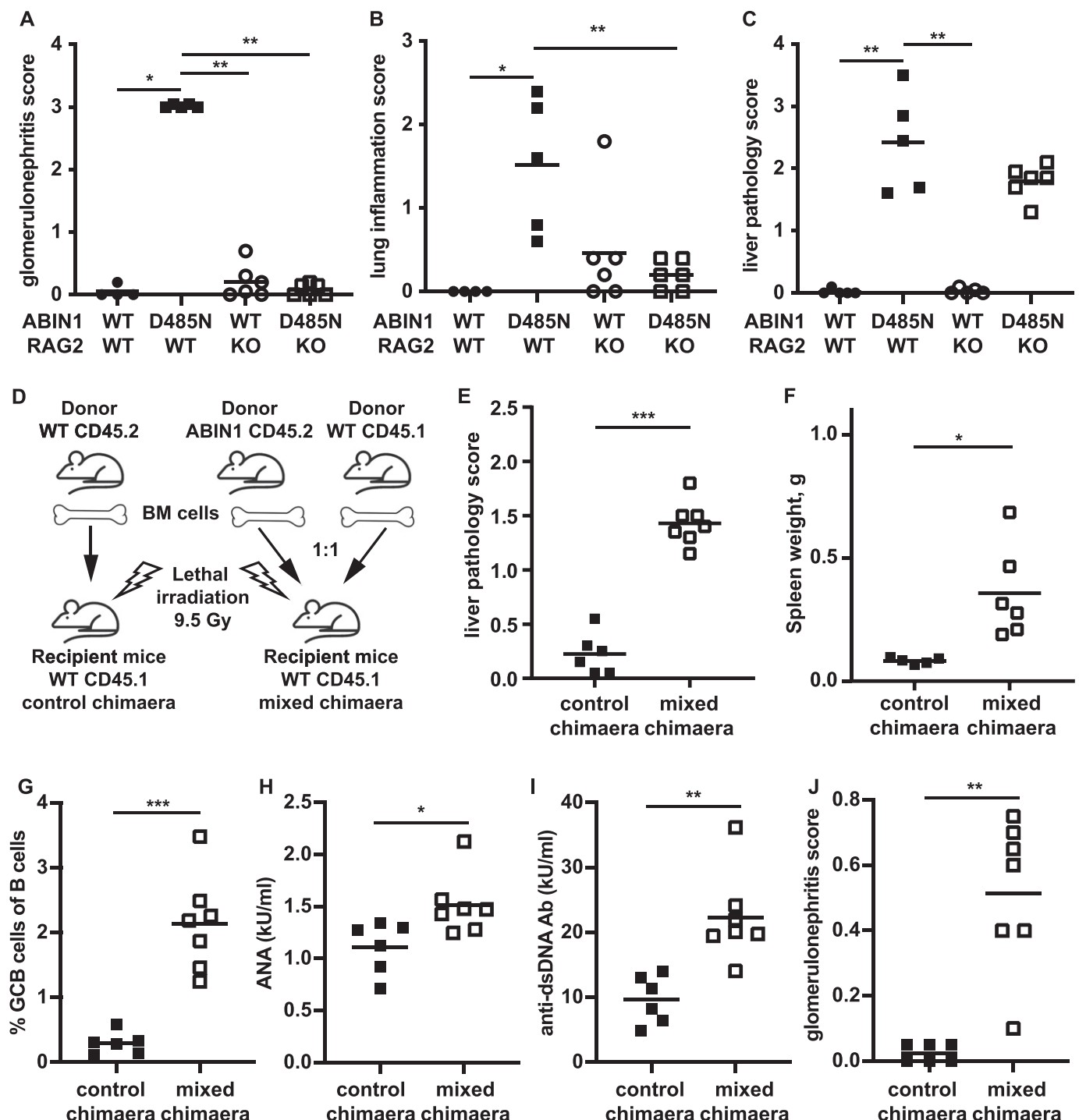

**Figure 2.  Glomerulonephritis but not liver pathology is independent of the adaptive immune system in ABIN1[D485N] mice and requires one or more myeloid cell types.**
**(A, B, C)** glomerulonephritis (A), lung inflammation (B), and liver pathology (C) from 24 to 26-wk-old WT, ABIN1[D485N], RAG2 KO, and ABIN1[D485N] × RAG2 KO mice (4–6 mice of each genotype). **(D)** Schematic showing how the bone marrow chimaera was generated. **(E, F, G, H, I, J)** show Liver pathology score (E), spleen weights (F), splenic GCB cells (B220[+ve] CD95[+ve]GL-7[+ve]) as a % of the total B220[+ve] B cells (G), ANA (H) and anti-dsDNA antibodies (I), and glomerulonephritis scores (J) from control and mixed chimaeric mice (5–7 mice of each experimental group). **(A, B, C, E, F, G, H, I, J)** Each symbol shows the data from a single mouse. Significance of the difference between the two groups was calculated using the unpaired *t* test with Welch's correction (E, F, I, J), or the Mann–Whitney test (A, B, C, G); * denotes $P < 0.05$, ** denotes $P < 0.01$, and *** denotes $P < 0.001$.
Source data are available for this figure.

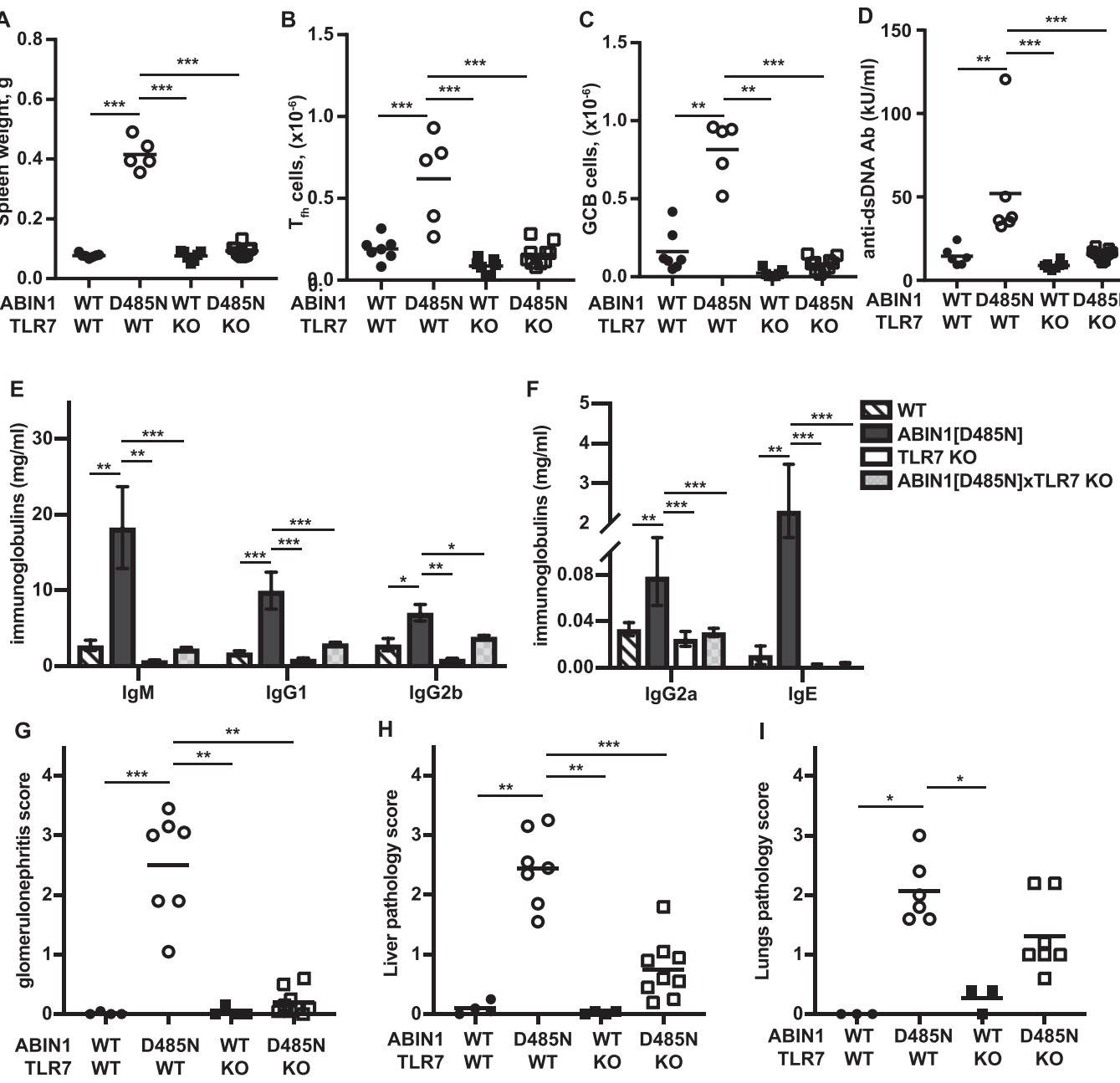

**Figure 3. Differences in autoimmunity, glomerulonephritis, liver, and lung pathology in ABIN1[D485N] × TLR7 KO mice.**
**(A)** Spleen weights of 17-wk-old WT mice (n = 6), ABIN1[D485N] mice (n = 5), TLR7 KO mice (n = 7), and ABIN1[D485N] × TLR7 KO (n = 10) mice. **(B, C)** Numbers of $T_{fh}$ (B) and GCB (C) cells in the spleens of 17-wk-old WT (n = 7), ABIN1[D485N] (n = 5), TLR7 KO (n = 4), and ABIN1[D485N] × TLR7 KO mice (n = 8). **(D, E, F)** Concentrations of different antibodies in the serum of 26-wk-old WT (n = 6), ABIN1[D485N] (n = 6), TLR7 KO (n = 8), and ABIN1[D485N] × TLR7 KO mice (n = 11). **(E, F)** The error bars in (E) and (F) are shown ± SEM. **(G, H, I)** Kidney (G), liver (H), and lung (I) pathology scores from WT, ABIN1[D485N], TLR7 KO, and ABIN1[D485N] × TLR7 KO mice (4–8 mice of each genotype). **(A, B, C, D, G, H, I)** Each symbol represents a biological replicate from one mouse. In A, B, E-IgG1 and F-IgG2a the significance between genotypes was calculated using one-way ANOVA and Tukey's post hoc test; significance in (C, D, E-IgM and IgG2b, G, H, I) was calculated using the Kruskal–Wallis nonparametric multiple comparison and the Mann–Whitney test; * denotes $P < 0.05$, ** denotes $P < 0.01$, and *** denotes $P < 0.001$.
Source data are available for this figure.

## Increased numbers of patrolling and inflammatory monocytes in the blood and organs of ABIN1[D485N] mice

To identify which myeloid cells might contribute to the disease phenotype of ABIN1[D485N] mice, we characterised the myeloid cell populations in different tissues (Figs 4 and 5). These studies revealed greatly increased numbers of patrolling monocytes (pMo) in the blood of ABIN1[D485N] mice (CD11b$^{+ve}$CD115$^{+ve}$Ly6C$^{-ve}$CX$_3$CR1$^{+ve}$MHCII$^{-ve}$), and modestly increased numbers of inflammatory monocytes (iMo, defined as CD11b$^{+ve}$CD115$^{+ve}$Ly6C$^{+ve}$CX$_3$CR1$^{+ve}$) (Figs 4A and B) (Geissmann et al, 2003; Auffray et al, 2007). There were also increased numbers of both pMo and iMo in the spleen (Fig 4C), where an alternative

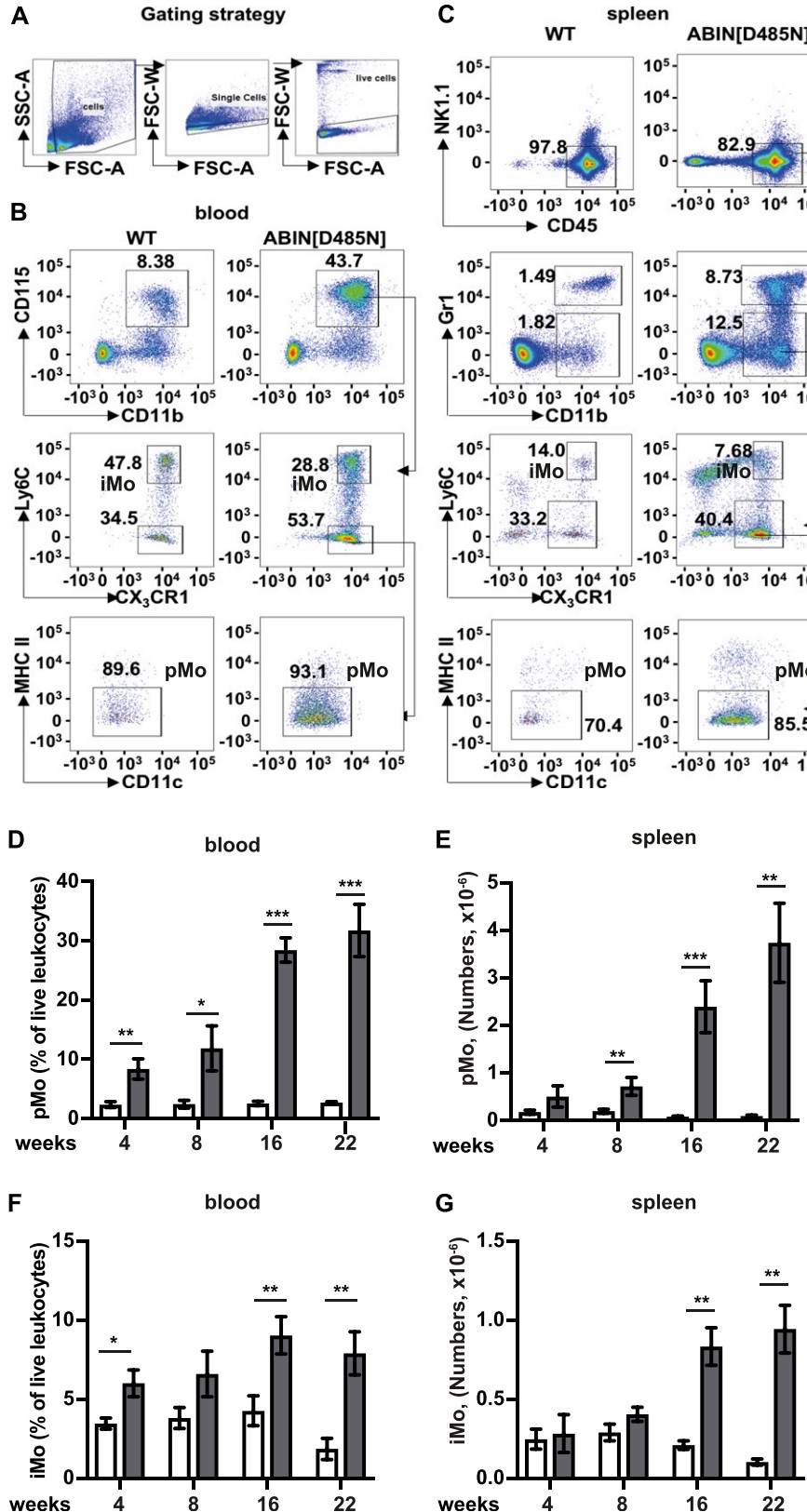

**Figure 4. Characterisation of the monocyte populations in the blood and spleen of ABIN1 [D485N] mice.**
**(A)** The cells were gated based on Forward Scatter - Area (FSC-A) and Side Scatter - Area (SSC-A) followed by exclusion of doublets and dead cells. **(B, C)** Flow cytometry characterization of monocytes in the blood (B) and spleen (C) of 22-wk-old mice. Patrolling monocytes (pMo) in the blood were characterised as CD115$^{+ve}$CD11b$^{+ve}$Ly6C$^{-ve}$CX3CR1$^{+ve}$MHCII$^{-ve}$ and inflammatory monocytes (iMo) as CD115$^{+ve}$CD11b$^{+ve}$Ly6C$^{+ve}$CX3CR1$^{+ve}$. Splenic pMo were identified as CD45$^{+ve}$Gr-1$^{-ve/low}$NK1.1$^{-ve}$CD11b$^{+ve}$Ly6C$^{+ve}$CX3CR1$^{+ve}$MHCII$^{-ve}$ and splenic iMo as CD45$^{+ve}$Gr-1$^{-ve/low}$NK1.1$^{-ve}$CD11b$^{+ve}$Ly6C$^{+ve}$CX$_3$CR1$^{+ve}$. **(D, F)** pMo (D) and iMo (F) are shown as a % of the live leukocytes in the blood of WT mice (white bars; n = 6–10) and ABIN1[D485N] mice (grey bars; n = 6–8). **(E, G)** Same as (D) and (F), except that the total numbers of pMo and iMo in the spleen were measured. The error bars show ± SEM. **(D, E, F, G)** Significance between the genotypes at each time points was calculated using the unpaired *t* test with Welch's correction in (D, F) or the Mann–Whitney test (E, G); *denotes $P < 0.05$, ** denotes $P < 0.01$, and *** denotes $P < 0.001$. Source data are available for this figure.

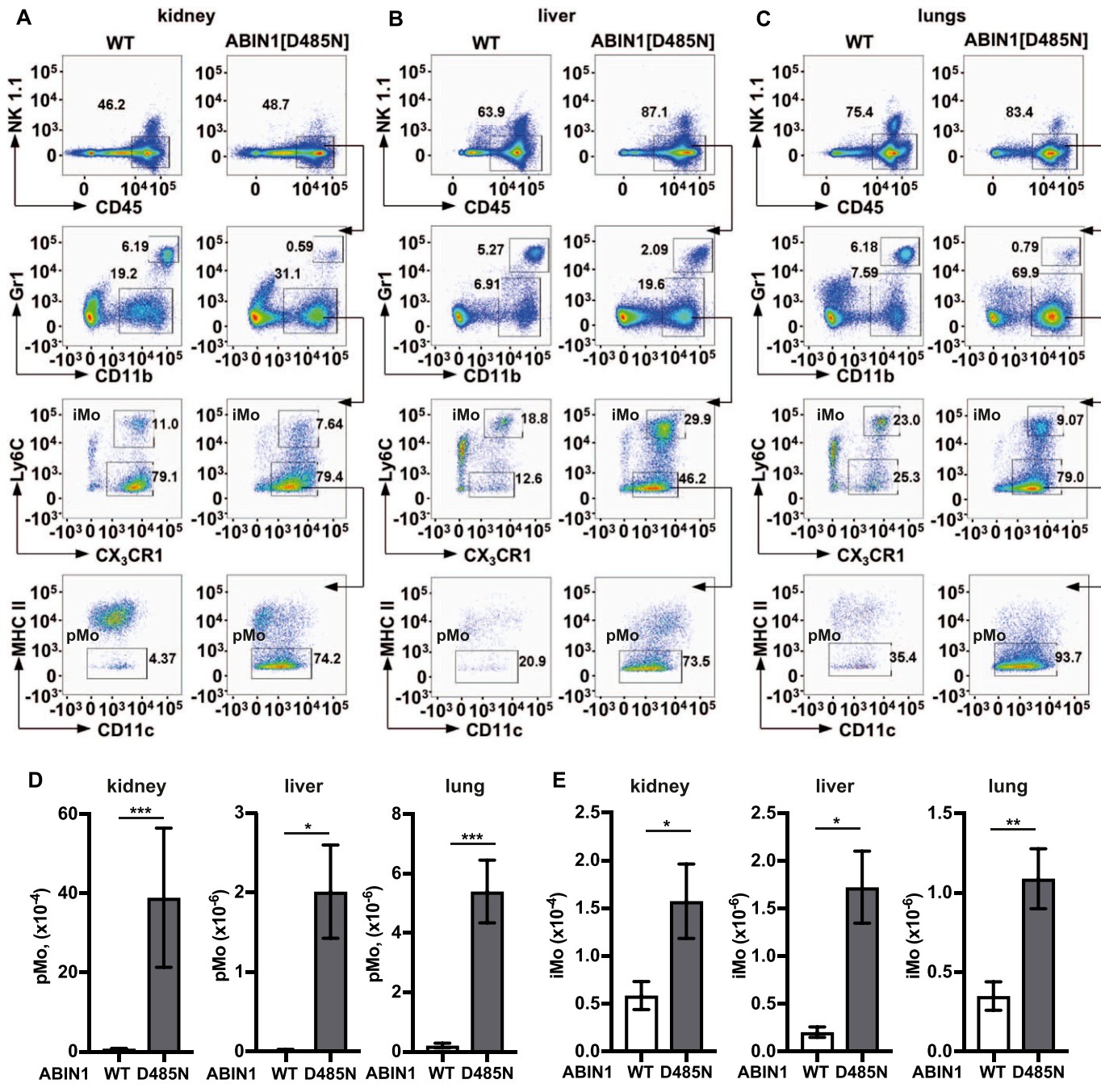

**Figure 5. Characterisation of the monocyte populations in the kidney, liver, and lungs of ABIN1[D485N] mice.**
**(A, B, C)** Same as (C) in Fig 4, except that the pMo and iMo in the kidney (A), liver (B), and lungs (C) were analysed. **(D, E)** Numbers of pMo (D) and iMo (E) in the kidney, liver, and lungs of 20–22-wk-old WT mice (white bars; n = 6) and ABIN1[D485N] mice (grey bars; n = 6). The error bars show ± SEM. **(D, E)** Significance between the genotypes was calculated using the unpaired *t* test with Welch's correction in (D) (liver) and (E) (liver and kidney) or the Mann–Whitney test in (D) (kidney and lung) and (E) (lung); * denotes *P* < 0.05 and ** denotes *P* < 0.01.
Source data are available for this figure.

gating strategy was necessary since the stability, and hence staining of CD115 is reduced when these organs are processed (Breslin et al, 2013). The pMo in spleen were defined as CD45$^{+ve}$Gr-1$^{-ve/low}$ NK1.1$^{-ve}$CD11b$^{+ve}$Ly6C$^{-ve}$CX$_3$CR1$^{+ve}$MHCII$^{-ve}$ and the iMo as CD45$^{+ve}$Gr-1$^{-ve/low}$NK1.1$^{-ve}$CD11b$^{+ve}$ Ly6C$^{+ve}$ CX$_3$CR1$^{+ve}$ (Fig 4C) (Jung et al, 2000).

Remarkably, increased numbers of pMo in the blood and spleen were already detectable when the ABIN1[D485N] mice were only 4 wk old, 8 wk before the onset of splenomegaly and increase in ANA (Figs 4 and S4) and 12 wk before glomerulonephritis, liver pathology, or lung inflammation became detectable (Nanda et al, 2011; Caster et al, 2013). The increase in pMo numbers became even more striking

as the mice aged, the blood pMo population reaching 29 and 33% of the total number of live leukocytes after 16 and 22 wk of age, respectively (Fig 4D). The number of pMo in the spleen also increased strikingly as the mice aged (Fig 4E), whereas increases in iMo in the blood (Fig 4F) and spleen (Fig 4G) were more modest. There was also a striking increase in the pMo and iMo present in the kidney, liver, and lungs of the ABIN1[D485N] mice (Figs 5A–E).

### Infiltration of different immune cells into the organs of ABIN1[D485N] mice

In view of the remarkable increase in patrolling monocytes in ABIN1[D485N] mice, we also examined the infiltration of other immune cells into the kidney, liver, and lungs of 20-wk-old mice (Figs S5 and S6). We observed increased numbers of neutrophils in the livers of ABIN1[D485N] mice compared with WT mice, but decreased numbers of neutrophils in the kidney and lungs (Fig S5A), and the potential significance of these observations is discussed later. There were also marked increases in the number of monocyte-derived dendritic cells (MoDCs) in the liver and lungs, but not in the kidney (Fig S5B). The eosinophil numbers in the kidneys, liver, and lungs (Fig S5C) and NK cell numbers in the kidneys and lungs (Fig S5D) were decreased in ABIN1[D485N] mice, whereas macrophage numbers in the three organs did not differ significantly in WT and ABIN1[D485N] mice (Fig S5E).

The total number of CD4$^+$ and CD8$^+$ T cells was increased in the kidney and liver, but not in the lungs (Figs S6A and B), but the percentage of activated (CD44$^{+ve}$) T cells was higher in all three organs in ABIN1[D485N] mice (Figs S6C and D). Finally, B cell numbers in the kidney and lungs of WT and ABIN1[D485N] mice were similar, but increased in the liver of ABIN1[D485N] mice (Fig S6E).

### The MyD88-IRAK4-IRAK1 signaling axis drives the increase in patrolling and inflammatory monocytes in ABIN1[D485N] mice by an IL-6–independent mechanism that does not require the adaptive immune system

To investigate which signaling components were responsible for increasing the number of pMo and iMo, we crossed the ABIN1[D485N] mice to TLR7 KO mice and MyD88 KO mice and to mice expressing kinase-inactive mutants of IRAK4 (IRAK4[D329A]) or IRAK1 (IRAK1[D359A]). Each cross strongly suppressed the numbers of pMo and iMo in the blood and spleen, apart from the cross to TLR7 KO mice, where the decrease in iMo numbers in the blood was modest (Figs 6A–D). In contrast, the pMo and iMo numbers were not affected, or only affected slightly, by crossing ABIN1[D485N] mice to RAG2 KO or IL-6 KO mice, respectively (Figs 6E–L). The increased numbers of pMo and iMo in ABIN1[D485N] mice are, therefore, driven by the same MyD88-IRAK4-IRAK1 signaling pathway that drives lupus but, similar to the liver pathology, the increases in these monocytes are unaffected by the absence of IL-6 or an adaptive immune system.

### The gene expression profiles of splenic pMo, iMo, and neutrophils are strikingly different in ABIN1[D485N] and WT mice

To characterize the pMo and iMo in greater detail, we performed RNA sequencing on the purified splenic monocytes. The gene expression profiles of the pMo and iMo from ABIN1[D485N] and WT mice show many similarities, but also marked differences (Figs 7A and B, and S7A and B). The up-regulated mRNAs in ABIN1[D485N] monocytes included that encoding myeloperoxidase (MPO) and, consistent with this finding, the MPO protein was also up-regulated (Figs S7C and D, and S8A). MPO catalyses the formation of hypochlorous acid (Hampton et al, 1998), a potent bactericidal nonradical oxidant. The RNAs encoding the proteinases elastase (*elane*), proteinase 3 (*prtn3*), cathepsin G (*ctsg*), and cathelicidin (*camp*) were also up-regulated in pMo and iMo from ABIN1[D485N] mice. These mRNAs, as well as those encoding MPO, the TLR4 ligands S100a8 and S100a9, and lactotransferrin (*ltf)*, are also up-regulated in splenic neutrophils from ABIN1[D485N] mice (Figs 7C and S8A–H, and Table S1). The potential significance of these findings is discussed later.

Principal component analysis showed that the gene transcription profiles of the splenic iMo from ABIN1[D485N] mice differed markedly from WT iMo and from both WT and ABIN1[D485N] pMo. A major difference was the high level of mRNAs encoding proteins required for cell division in ABIN1[D485N] iMo (Fig 7D and Table S2). Consistent with these findings, cell cycle analysis of the splenic iMo revealed that, in contrast to WT iMo, many ABIN1[D485N] iMo were in the S and G2 phases of the cell division cycle (Figs 7E and F). This may account for increased iMo numbers and the even larger increase in the pMo numbers in ABIN1[D485N] mice, because the pMo are derived from iMo (see the Discussion section).

### All facets of the disease phenotype of ABIN1[D485N] mice are reduced by an orally active IRAK4 inhibitor

Our earlier studies had shown that autoimmunity and glomerulonephritis in ABIN1[D485N] mice were prevented by crossing to MyD88 KO mice (Nanda et al, 2011) or mice expressing kinase-inactive mutants of IRAK4 or IRAK1 (Nanda et al, 2016). These observations and the critical role of the MyD88-IRAK4-IRAK1 signaling axis in all aspects of the disease phenotype led us to study the effects of PF 06426779, an orally active IRAK4-specific inhibitor (Lee et al, 2017) (Fig S9A). The inclusion of this compound in the food of adult WT mice (4g/kg food) caused the level of PF06426779 in the serum to oscillate over the 12 h/12 h light/dark cycle, peaking at 6 $\mu$M in the dark cycle when the mice were feeding and declining to 1.5 $\mu$M during the light cycle, when they were not.

The compound was given to ABIN1[D485N] mice for 10 wk, starting at 6–8 wk of age. At this age, increased numbers of pMo and iMo were already present, but splenomegaly, auto-immunity, and organ pathology were not (Nanda et al, 2011). We found that PF 06426779 prevented further increases in pMo numbers in the blood (Fig 8A), as well as splenomegaly (Fig 8B) and the formation of splenic GCB cells (Fig 8C). The IRAK4 inhibitor also reduced the concentration of anti-dsDNA (Fig 8D) and attenuated glomerulonephritis (Figs 8E and S9B). Importantly, the IRAK4 inhibitor also reduced liver inflammation (Figs 8F and S9C), and there was a statistically significant decrease in lung inflammation (Figs 8G and S9D). In contrast, the pMo in the blood of control ABIN1[D485N] mice not given the IRAK4 inhibitor continued to increase, and these mice developed splenomegaly, auto-antibodies, glomerulonephritis, liver pathology, and lung inflammation (Figs 8A–G).

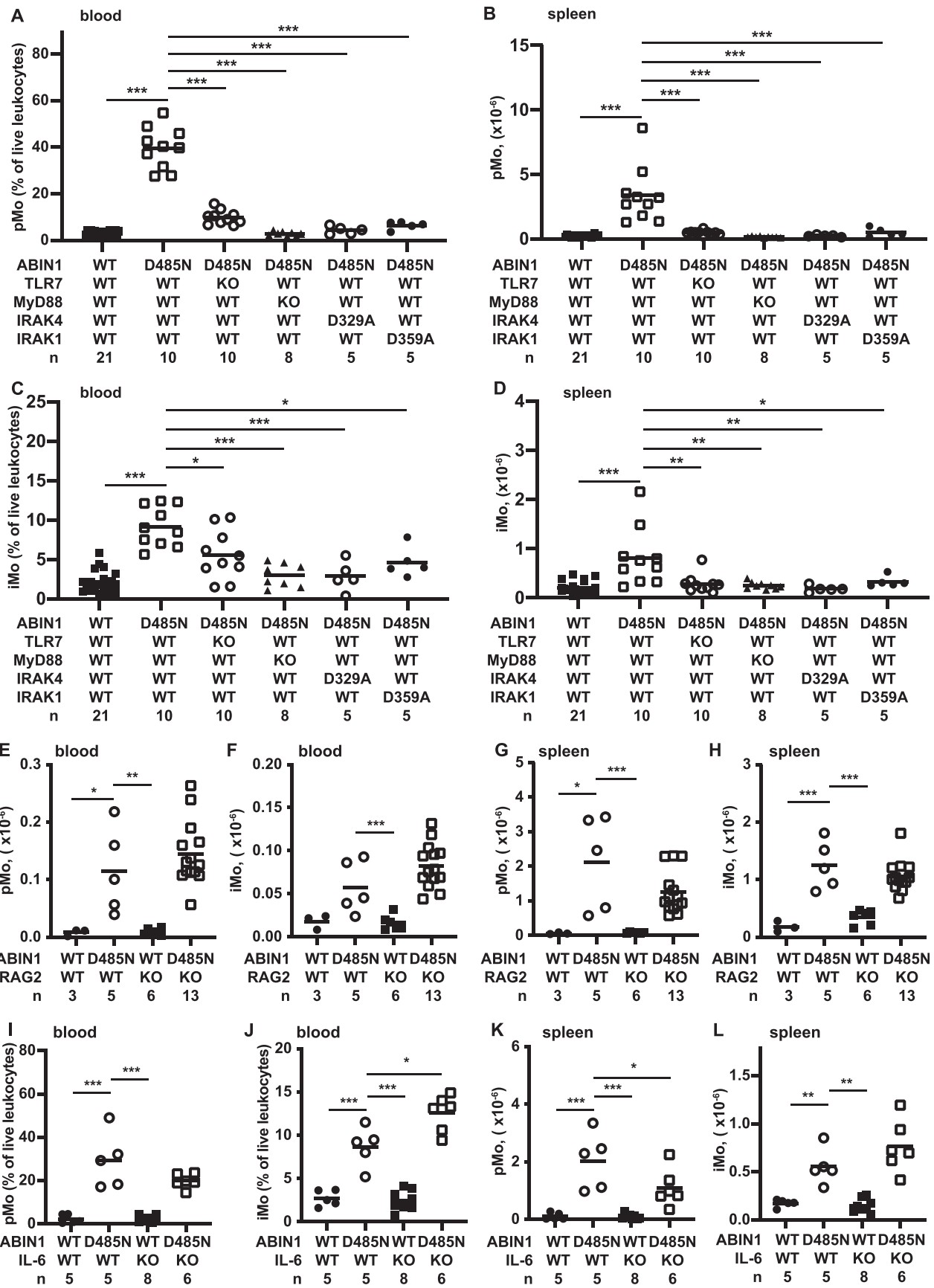

Mechanism of lupus in ABIN1[D485N] mice   Nanda et al.   https://doi.org/10.26508/lsa.201900533   vol 2 | no 6 | e201900533   9 of 19

### An IRAK4 inhibitor reduces cytokine secretion by monocytes

The stimulation of splenic pMo and iMo from ABIN1[D485N] mice with R837, a ligand for TLR7, induced the secretion of TNF and the chemokines CCL3 and CCL4 (Figs 8H–M). The iMo additionally secreted IL-6 and IL-12p40 (Figs 8N and O). The IRAK4 inhibitor PF0642779 reduced R837-stimulated cytokine secretion from pMo and iMo (Figs 8H–O), indicating that IRAK4 catalytic activity not only drives the increase in pMo numbers in ABIN1[D485N] mice but also the secretion of cytokines by these cells. The stimulation of iMo with LPS induced the secretion of IL-6, CCL5, CXCL1, and G-CSF, which was also suppressed by PF06426779 (Figs 8 and S9E).

## Discussion

Studies aimed at gaining a molecular understanding of the pathogenetic mechanisms underlying lupus have focused mainly on autoimmunity and how it triggers glomerulonephritis in this disease. Here, we also examined the liver pathology and lung inflammation in ABIN1[D485N] mice, where lupus arises spontaneously from a point mutation that disables the ubiquitin-binding function of a negative regulator of TLR-MyD88-IRAK4-IRAK1 signaling. We found that the expression of IL-6 in ABIN1[D485N] mice was needed for the development of autoimmunity culminating in glomerulonephritis, but had no effect on liver pathology or lung inflammation (Fig 1). Moreover, in contrast to glomerulonephritis, the development of the liver pathology did not require the adaptive immune system (Fig 2) and instead was driven by non-B, non-T cells. These observations indicate that the disease phenotype arises from a combination of autoimmunity and auto-inflammation.

Autoimmunity in ABIN1[D485N] mice is clearly dependent on one or more endogenous TLR7-activating ligands (Fig 3), presumably an RNA(s) released during cell death and/or produced by commensal microbiota. The latter possibility implies that the identity of the ligand(s) may vary with the composition of the microbiome and could, therefore, vary with the facility in which the mice are housed. Recently, another laboratory reported that crossing ABIN1 KO mice to either TLR7 KO mice or TLR9 KO mice had little effect on autoimmunity or glomerulonephritis and that crossing to TLR7/TLR9 double KO mice was required to prevent these hallmarks of lupus (Kuriakose et al, 2019). This is unexpected because the disease phenotype of other lupus-prone mouse strains is prevented by crossing to TLR7 KO mice, whereas crossing to TLR9 KO mice has actually been reported to exacerbate lupus by causing the up-regulation of TLR7 (Christensen et al, 2006; Nickerson et al, 2010; Stoehr et al, 2011). Other differences in the mechanisms that negatively regulate TLR7 and TLR9 signaling have recently been identified (Majer et al, 2019). The difference

between the findings of Kuriakose et al and those reported here might be explained by the distinct microbiomes of these mice and/or by the loss of additional functions of the ABIN1 protein in the ABIN1 KO mice. Only the ubiquitin-binding function of the protein is disabled in ABIN1[D485N] mice, but the interaction of ABIN1 with A20 (the product of the *tnfaip3* gene) and other, as yet unidentified, functions of ABIN1 will also be lost in ABIN1 KO mice. The loss of additional functions might also explain why ABIN1 KO mice start dying when they are only 4 wk old. Indeed, 50% of the ABIN1 KO mice had died after 8 wk, before glomerulonephritis was detectable (Kuriakose et al, 2019), suggesting that the early death is unrelated to glomerulonephritis. In contrast, ABIN1[D485N] mice only start dying when severe multiorgan damage has developed at 6 mo of age (Nanda et al, 2011). The partial alleviation of liver pathology and lack of a significant reduction in lung inflammation in ABIN1[D485N] × TLR7 KO mice suggests the involvement of an additional TLR(s) and its endogenous ligand(s) in the development of liver and lung pathology. The ABIN1 KO mice also display liver and lung inflammation (Zhou et al, 2011), but quantitative scoring and the effect of crossing to TLR7 KO and/or TLR9 KO mice on these disease endpoints was not reported (Kuriakose et al, 2019).

An inference from our results is that anti-IL-6 antibodies or TLR7 antagonists may be ineffective, or only partially effective, in preventing lupus, and the failure of an anti-IL-6 monoclonal antibody to achieve its primary end goals in a phase 2 clinical trial for lupus (Wallace et al, 2017) is consistent with this notion. These considerations suggest that only drugs targeting a core component of the MyD88-IRAK4-IRAK1 pathway (Fig 9) will have the potential to suppress all facets of the lupus phenotype and led us to treat the ABIN1[D485N] mice with an orally active IRAK4 inhibitor. The results demonstrated that the IRAK4 inhibitor did indeed reduce both the autoimmune and autoinflammatory aspects of the disease (Fig 8).

A different IRAK4 inhibitor (BMS-986126) has been shown to reduce the level of anti-dsDNA in the serum and to suppress kidney pathology in two other lupus-prone mouse lines (Murphy Roths Large/ lymphoproliferation (MRL/lpr) and New Zealand Black/ White (NZB/W) mice) (Dudhgaonkar et al, 2017), but its efficacy in alleviating the pathology and inflammation of other organs was not reported. Lupus in MRL/lpr mice seems to be caused by a mutation in the CD95 receptor that prevents the removal of autoreactive T and B cells, whereas the mutation(s) causing lupus in NZB/W mice is (are) unknown. However, autoimmunity and kidney pathology were both attenuated by crossing MRL/lpr mice to TLR7 KO mice (Christensen et al, 2006) or delayed by the administration of TLR7 antagonists to NZB/W mice (Dong et al, 2005). These observations, together with the efficacy of an IRAK4 inhibitor in reducing autoimmunity imply that TLR7-MyD88-IRAK4-IRAK1 signaling is up-regulated in MRL/lpr and NZB/W mice, but the mechanism(s) causing hyperactivation in these mouse strains is (are) unclear.

**Figure 6. The pMo and iMo in the blood and spleen of different mouse lines.**
**(A, B, C, D, E, F, G, H, I, J, K, L)** ABIN1[D485N] mice were crossed to TLR7 KO, MyD88 KO, IRAK4[D329A], or IRAK1[D359A] mice (A, B, C, D), RAG2 KO mice (E, F, G, H) or IL-6 KO mice (I, J, K, L). **(A, B, C, D, E, F, G, H, I, J, K, L)** The pMo in the blood (A, E, I), and spleen (B, G, K) and iMo in the blood (C, F, J) and spleen (D, H, L) were measured in 17-wk-old mice with the indicated genotypes. The numbers of mice analysed are denoted by n and each symbol represents an individual mouse. Statistical significance between the genotypes is shown by the horizontal bars and asterisks and was calculated using one-way ANOVA and the Tukeys post hoc test (A, B, C, H, I, J, K, L) or the Kruskal–Wallis and the Mann–Whitney tests (D, E, F, G); * denotes $P < 0.05$, ** $P < 0.01$, and *** denotes $P < 0.001$. No horizontal bars are shown for data sets that have not reached statistical significance.
Source data are available for this figure.

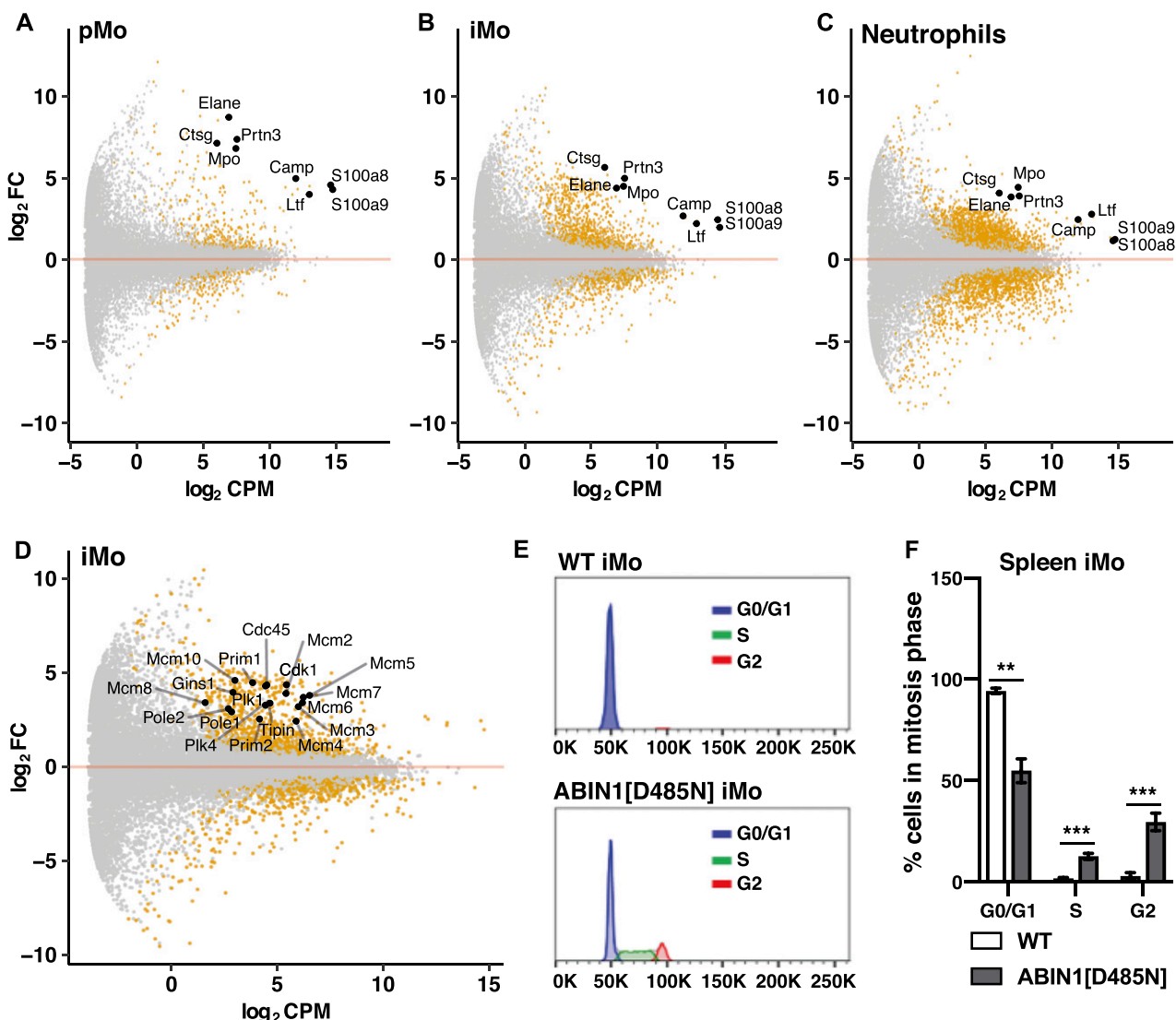

**Figure 7. RNA-seq of splenic monocytes and neutrophils in ABIN1[D485N] mice.**
**(A, B, C)** MA (M - log ratio and A - mean average) plots showing differential expression of selected genes in pMo (A), iMo (B), and neutrophils (C) obtained by RNA-seq analysis of cells purified from the spleens of 20–22-wk-old WT (n = 4) and ABIN1[D485N] mice (n = 4). Significantly up-regulated or down-regulated genes are shown in yellow. The Y-axis shows $\log_2$ fold change and X-axis shows $\log_2$ of normalized gene expression (counts per million [cpm]) in the ABIN1[D485N] cells. (D) As in B, except that up-regulated genes involved in regulating cell division in ABIN1[D485N] iMo are shown. **(E, F)** Representative flow cytometry plots (E) and bar graph (F) showing cell cycle analysis of iMo in the spleens of 14-w-old WT (n = 4) and ABIN1[D485N] (n = 3) mice. **(F)** Statistical significance between the genotypes was calculated using two-way ANOVA and the Sidak post hoc test (F); *** denotes $P < 0.001$.
Source data are available for this figure.

Polymorphisms in four genes encoding proteins that control the MyD88-IRAK4-IRAK1 pathway (Fig 9) have been reported to pre-dispose to SLE and other autoimmune diseases in many human populations, namely, *TLR7* (Shen et al, 2010; Tian et al, 2012; Lee et al, 2016), *IRAK1* (Jacob et al, 2007, 2009; Kaufman et al, 2013), *TNIP1* (Gateva et al, 2009; Han et al, 2009; Nair et al, 2009; Adrianto et al, 2012; Gregersen et al, 2012; Shi et al, 2014), and *TNFAIP3* (Musone et al, 2008; Bates et al, 2009; Adrianto et al, 2011). These observations suggest that hyperactivation of the MyD88-IRAK4-IRAK1 pathway contributes to disease progression in a significant proportion of human lupus patients. Lupus is far more prevalent in women than men (Tedeschi et al, 2013) and it is, therefore, of interest that the

*TLR7* and *IRAK1* genes are X-linked in both humans and mice. Importantly, *TLR7* escapes X-chromosome inactivation in human immune cells, and the TLR7 protein is expressed in the myeloid cells of women at twice the level present in men (Souyris et al, 2018). This may contribute to the prevalence of lupus in women; recall that duplication of the *tlr7* gene in mice causes lupus to develop spontaneously (Pisitkun et al, 2006).

The two types of monocytes present in mice have been termed inflammatory (iMo) and patrolling (pMo), the former giving rise to the latter. The iMo also give rise to macrophages and monocyte-derived dendritic cells (reviewed Ginhoux & Jung (2014)). Previous studies reported high levels of monocytes in at least two lupus-prone

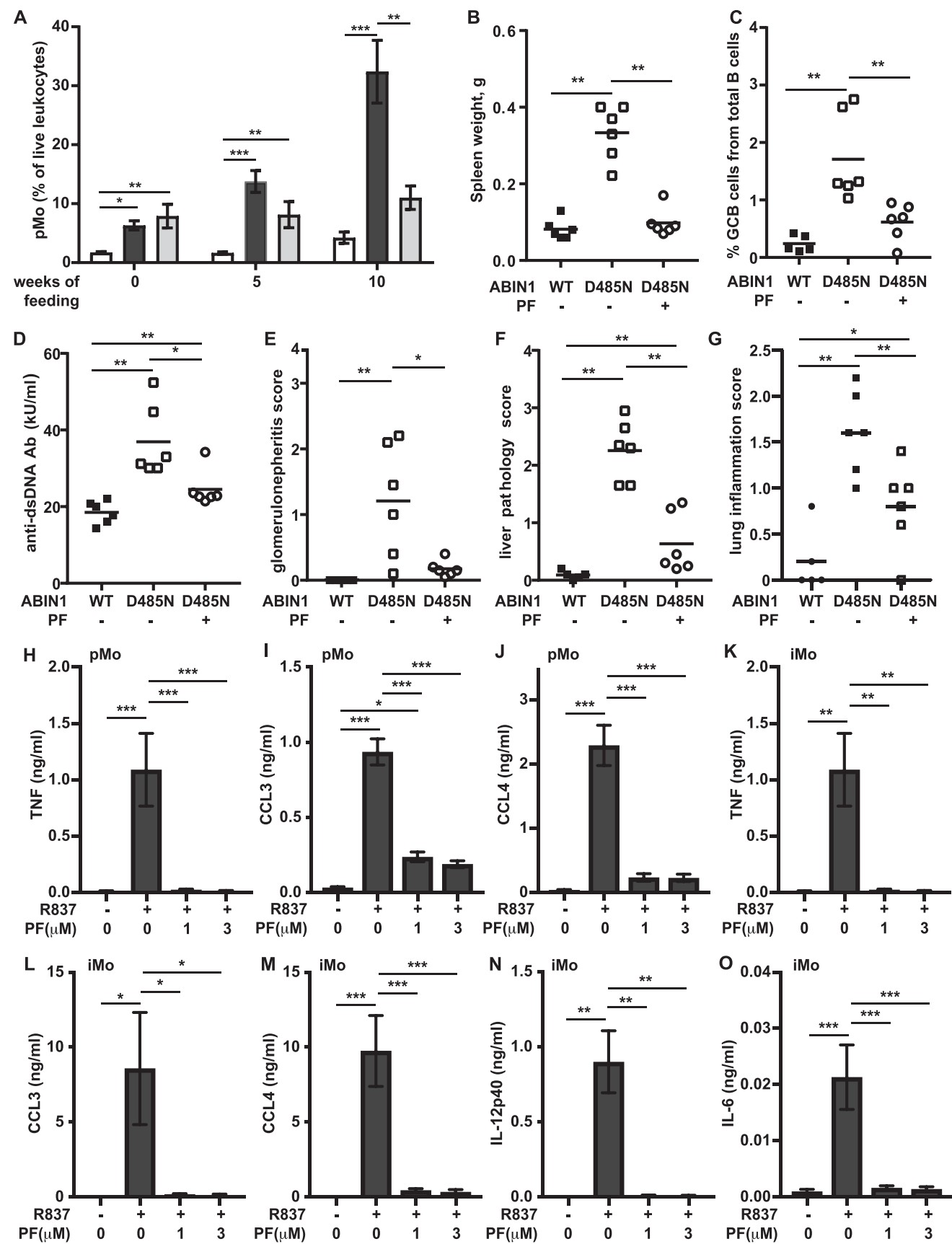

mouse strains (Wofsy et al, 1984; Santiago-Raber et al, 2009a), whereas high levels of nonclassical monocytes, thought to be equivalent to pMo in mice, were reported in human lupus patients (Cros et al, 2010; Kuriakose et al, 2019). Here, we studied each type of monocyte individually and found particularly high levels of pMo in the blood, spleen, and other organs of ABIN1[D485N] mice (Figs 4 and 5). High levels of pMo in the blood and kidneys of ABIN1 KO mice were also reported recently (Kuriakose et al, 2019). Here, we made the novel and potentially important finding that the number of pMo in the blood of ABIN1[D485N] mice are already increased considerably months before autoimmunity or organ pathology became detectable (Figs 4D and E). If this is also the situation in humans, then the measurement of pMo numbers in the blood might be a useful diagnostic indication of those at risk of developing lupus, such as the siblings of lupus patients or humans with polymorphisms in genes known to predispose to lupus, allowing early intervention with IRAK4 and/or IRAK1 inhibitors to prevent the onset of the disease.

Mice with a conditional deletion of ABIN1 in myeloid cells were recently reported to develop autoantibodies and glomerulone-phritis, indicating an essential role for myeloid cells in driving autoimmunity in ABIN1 KO mice (Kuriakose et al, 2019). These observations can be rationalized with our findings (Fig 1) if it is myeloid cells that produce the IL-6 needed to drive autoimmunity and glomerulonephritis. The iMo from ABIN1[D485N] mice secreted IL-6 in response to TLR ligands (Fig 8O) and could be one source of the IL-6. The splenic pMo did not secrete IL-6 in response to these stimuli, but chemokines secreted by pMo may recruit other IL-6–secreting cells to the organs that they infiltrate. The conventional dendritic cells and B cells of ABIN1[D485N] mice also overproduce IL-6 when stimulated with TLR-activating ligands (Nanda et al, 2011).

Glomerulonephritis in ABIN1[D485N] × RAG2 KO mice was un-detectable up to 6 mo of age, indicating that a functional adaptive immune system is required for this phenotype to develop. The pMo numbers in the blood and spleen of ABIN1[D485N] × RAG2 KO mice were similar to those present in ABIN1[D485N] mice (Figs 6E and G), indicating that the pMo numbers in ABIN1[D485N] mice increase independently of the adaptive immune system.

There was also a striking increase in the infiltration of pMo into the kidneys, liver, and lungs of ABIN1[D485N] mice (Figs 5D and E), suggesting that pMo may contribute to the pathology observed in these organs. Further evidence that pMo contribute to the kidney pathology is suggested by the recent finding that chimaeric mice devoid of pMo, which were generated by injecting liver cells from the foetus of ABIN1 KO × Nr4a1 KO mice into irradiated WT mice, did not develop glomerulonephritis (Kuriakose et al, 2019). Surprisingly,

and in contrast to our findings in ABIN1[D485N] × RAG2 KO mice, the ABIN1 KO × RAG1 KO mice, which also lack T and B cells, still developed glomerulonephritis (Kuriakose et al, 2019). The reason for this difference with our results is unclear, although it might be related to the more severe phenotype of ABIN1 KO mice.

Crossing the ABIN1[D485N] mice to RAG2 KO mice not only abolished glomerulonephritis but also lung inflammation, indicating that the pathology of both of these organs requires the adaptive immune system. Consistent with these findings, there were increased numbers of T cells in the kidneys of ABIN1[D485N] mice and an increased proportion of activated T cells in both the kidneys and lungs (Fig S6). In contrast, despite increased numbers of T and B cells and a higher proportion of activated T cells in the liver, the liver pathology was unaffected in the ABIN1[D485N] × RAG2 KO mice. Thus, the liver pathology is driven by non-B, non-T cells, raising the question of which haematopoietic cells are responsible for the liver pathology.

Interestingly, we observed that the infiltration of neutrophils into the liver was increased in ABIN1[D485N] mice, whereas less neutrophils were present in the kidneys and lungs of ABIN1[D485N] mice than in WT mice (Fig S5A). An important function of neutrophils is their ability to undergo NETosis (neutrophil extracellular trap formation), a form of cell death that generates local high concentrations of antimicrobial compounds to kill pathogens extracellularly (Urban et al, 2009; Papayannopoulos et al, 2010; Aratani, 2018). Interestingly, increased NETosis has been linked to SLE (Hakkim et al, 2010) and MPO and proteinases released during NETosis cause tissue damage (Knight et al, 2012; Metzler et al, 2014; Bjornsdottir et al, 2015). It is, therefore, of great interest that the splenic neutrophils in ABIN1[D485N] mice expressed much higher levels of RNAs encoding proteins released during NETosis, suggesting that increased NETosis may be a factor driving the liver pathology. Moreover, RNA-seq analysis unexpectedly revealed that the pMo and iMo in ABIN1[D485N] mice also express high levels of RNAs encoding proteins important for NETosis (Fig 7 and Table S1). The numbers of pMo and iMo are greatly elevated in the livers of ABIN1[D485N] mice, and monocytes have been reported to undergo NETosis (Granger et al, 2017). Therefore, NETosis driven by neutrophils, pMo and iMo may all contribute to the liver pathology displayed by ABIN1[D485N] mice. However, there was also a striking increase in the number of monocyte-derived dendritic cells in the livers of ABIN1[D485N] mice (Fig S5). Therefore, increased production of molecules by MoDCs that can cause tissue damage, such as cytokines, may be another factor contributing to the liver pathology. The increased number of MoDCs could also contribute to T-cell activation and hence to the kidney and lung pathology.

**Figure 8.   The IRAK4 inhibitor PF 06426779 prevents autoimmunity and organ inflammation in ABIN1[D485N] mice.**

ABIN1[D485N] mice (6–8 wk old) or age-matched WT mice were fed for 10 wk on R&M3 diet without or with PF 06426779 (PF) (4 g/kg). **(A)** At the times indicated, pMo in the blood was quantitated. The WT mice and ABIN1[D485N] mice fed on normal R&M3 diet are shown by the white and dark grey bars, respectively, whereas the ABIN1[D485N] mice fed on R&M3 diet containing PF are shown by the light grey bars. **(B, C)** Spleen weights (B) and splenic GCB cells (B220$^{+ve}$ CD95$^{+ve}$GL-7$^{+ve}$) (C) were measured 10 wk after feeding WT (black squares) or ABIN1[D485N] mice (open symbols) with R&M3 diet without (−) or with (+) PF. **(D, E, F, G)** As in (C), except that anti-dsDNA antibodies in the serum (D), glomerulonephritis (E), liver pathology (F), and lung inflammation (G) were measured. **(B, C, D, E, F, G)** Each symbol represents data from a single mouse. **(H, I, J, K, L, M, N, O)** splenic pMo (5 × 10$^4$ cells) (H, I, J) and splenic iMo (5 × 10$^4$ cells) (K, L, M, N, O) from ABIN1[D485N] mice (n = 5) were incubated for 1 h with the indicated concentrations of PF, then stimulated for 6 h with 1.0 μg/ml R837. **(H, I, J, K, L, M, N, O)** TNF (H, K), CCL3 (I, L), CCL4 (J, M), IL-12p40 (N), and IL-6 (O) concentrations in the culture medium were then measured. The error bars show ± SEM. **(A, B, C, D, E, F, G, H, I, J, K, L, M, N, O)** Statistical significance between the experimental groups was calculated using the one-way ANOVA and the Tukey's post hoc test in (A, H, I, J, K, L, M, O) or the Kruskal–Wallis and the Mann–Whitney tests in (B, C, D, E, F, G, O); * denotes $P < 0.05$, **$P < 0.01$, and *** denotes $P < 0.001$.
Source data are available for this figure.

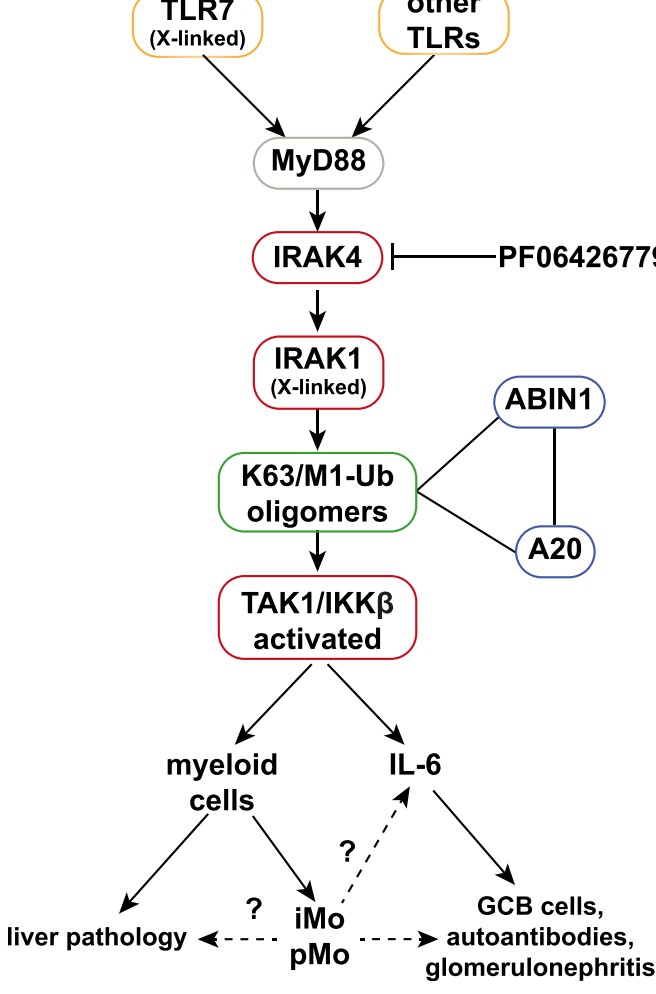

**Figure 9.  Pathways to autoimmunity and autoinflammation in ABIN1[D485N] mice.**
The MyD88-IRAK4-IRAK1 signaling pathway induces the formation of hybrid ubiquitin chains containing both Lys63 (K63)-linked and Met1 (M1)-linked ubiquitin (Ub) chains. ABIN1 and its binding partner A20 interact with these ubiquitin chains, restricting the activation of TAK1 and IKKβ. The ABIN1[D485N] mutation prevents interaction with ubiquitin chains causing hyperactivation of TAK1 and IKKβ and the overproduction of inflammatory mediators. The overproduction of IL-6 drives the formation of GCB cells, autoantibodies, and glomerulonephritis, but not the autoinflammatory aspects of the disease phenotype, which include the liver pathology. The liver pathology is driven by one or more myeloid cells independently of the adaptive immune system. The MyD88-IRAK4-IRAK1 signaling axis also drives increased formation of atypical patrolling and inflammatory monocytes (pMo, iMo) in ABIN1[D485N] mice which may contribute to the liver pathology, as well as IL-6 production and glomerulonephritis. The activation of TLR7 drives autoimmunity in ABIN1[D485N] mice, but additional TLRs contribute to other aspects of the disease phenotype. The IRAK4 inhibitor, PF06426779, suppresses both the autoimmune and autoinflammatory aspects of the disease phenotype of ABIN1[D485N] mice. Polymorphisms in TLR7, IRAK1, ABIN1, and A20 predispose to human lupus. The genes encoding TLR7 and IRAK1 are X-linked and may account, at least in part, for the prevalence of lupus in women.

Interestingly, and similar to our observations, a population of low-density gradient neutrophils present in human lupus patients had enhanced expression of the genes encoding MPO, elastase, and cathelicidin compared with neutrophils from healthy controls and

had a higher capacity to produce NETs and, hence, tissue damage (Villanueva et al, 2011; Carmona-Rivera et al, 2015).

Our genetic analysis established that hyperactivation of MyD88-IRAK4-IRAK1 signaling drives the increased numbers of pMo and iMo in ABIN1[D485N] mice (Fig 6) and, consequently, an IRAK4 inhibitor arrested the increase in pMo and iMo numbers in ABIN1[D485N] mice (Fig 8). The RNA-seq analysis of splenic iMo from ABIN1[D485N] mice revealed that they express far higher levels of mRNAs encoding proteins required for cell division than WT iMo (Fig 7D). Consistent with this finding, the splenic iMo from ABIN1[D485N] mice, but not WT mice, were found to be actively engaged in cell division (Figs 7E and F). Because iMo give rise to pMo (Yona et al, 2013), this finding can potentially explain the increased numbers of pMo in the blood (Fig 4D) and organs (Figs 4E, and 5D and E) of ABIN1[D485N] mice, as well as the increased numbers of iMo. Understanding how MyD88-IRAK4-IRAK1 signaling stimulates the transcription of genes controlling the cell division cycle in monocytes and the conversion of iMo to pMo will be interesting topics for future research.

## Materials and Methods

### Generation and maintenance of mouse lines

The ABIN1[D485N], ABIN1[D485N] × MyD88 KO, ABIN1[D485N] × IRAK4 [D329A], and ABIN1[D485N] × IRAK1[D359A] knock-in mice were generated as described (Nanda et al, 2011, 2016). The ABIN1[D485N] mice, back-crossed to C57Bl6/J mice (Charles River Laboratories) more than 15 generations, were crossed to TLR7 KO mice on a C57Bl6/J background (Jackson Laboratories). This C57Bl6/J strain is free of the DOCK2 mutation present in C57Bl mice from Harlan Laboratories, which affects the immune system (Mahajan et al, 2016). The C57Bl6/J CD45.1 mouse line was obtained from Charles River. Mice were provided with free access to food (R&M3 pelleted irradiated or autoclaved diet) and water. Animals were kept in individually ventilated cages at 21°C, 45–65% relative humidity and a 12 h/12 h light/dark cycle under specific-pathogen–free conditions in accordance with UK and European Union regulations. Experiments on mice were approved by the University of Dundee Ethical Review Committee under a UK Home Office project license.

### Isolation of cells and flow cytometry

The antibodies used for flow cytometry analysis and their sources are summarised in Table S3. Single-cell suspensions were made from the spleen and lungs (Nanda et al, 2011; Moro et al, 2015). The kidneys were cut into 1–2-mm segments and digested for 45 min at 37°C in 7 ml Roswell Park Memorial Institute (RPMI) supplemented with 50 μg/ml Liberase (Roche) and 10 μg/ml DNase I (Roche). Tissues were then further processed as described (Liao et al, 2017). For phenotypic analysis of the blood, 2–3 drops of blood were collected from mouse tails in ice-cold RPMI buffer containing 10% foetal bovine serum, 10% red blood cell lysis buffer (Sigma-Aldrich) and 2 mM EDTA or via cardiac puncture in MiniCollect EDTA tubes (Greiner Bio-One). Erythrocytes were eliminated by treatment for 2 min at 21°C with red blood cell lysis buffer. Cell suspensions were

stained with 0.5 $\mu$g/ml DAPI (BioLegend), acquired on a BD FACSVerse using BD FACSuite software (BD Bioscience) and total cell numbers calculated using FlowJo software (Tree Star, Inc.). For flow cytometry analysis, the cells were blocked for 20 min at 4°C with FcR antibody (purified anti-CD16/32; BD Pharmigen) diluted (1:50) in PBS containing 1% Bovine serum albumin. For detection of surface antigens, the cells were stained for 20 min at 4°C with the appropriate fluorophore-conjugated antibodies. To exclude dead cells, 0.5 $\mu$g/ml DAPI was added before analysis. To characterise MPO expression, splenic cells were first stained for surface marker expression and then fixed and permeabilized using Intracellular Fixation & Permeabilization Buffer Set according to the manufacturer's instructions. After permeabilization, the cells were stained for 1 h at 4°C with anti-MPO antibody (Abcam) and analysed. Data were collected using BD FACSCanto or BD LSRFortessa II and BD FACSDiva software (BD Bioscience) and the results analysed by FlowJo software. Doublets were excluded by gating for Forward Scatter-Area (FSC-A) and Forward Scatter-Width (FSC-W), whereas DAPI$^{-ve}$ cells were gated for further analysis to exclude dead cells.

### Cell cycle analysis

Splenocytes were stained for surface expression of CD11b and Ly6C, then washed, fixed for 30 min at 4°C in ice cold 70% ethanol, washed twice with PBS, and incubated for 5 min with RNase A (100 $\mu$g/ml) (Thermo Fisher Scientific). Propidium iodide (Sigma-Aldrich) was added to a final concentration of 50 $\mu$g/ml. Data were acquired using BD LSRFortessa II and analysed by FlowJo software.

### Cell sorting and cell stimulation

Splenocytes from ABIN1[D485N] mice were incubated FcR antibody as described earlier. The cells were then incubated for 20 min with anti-CD3, anti-CD19, anti-NK1.1, anti-Ly6G, and anti-Ter119 biotinylated antibodies. T, B, NK cells, neutrophils, and erythrocytes were depleted using Streptavidin MicroBeads and LD Columns (Miltenyi) according to Miltenyi's instruction. The cells from the flow-through were stained as described earlier. Patrolling monocytes (DAPI$^{-ve}$CD45$^{+ve}$CD11b$^{+ve}$NK1.1$^{-ve}$MHCII$^{-ve}$Ly6C$^{-ve}$CX$_3$CR1$^{+ve}$) and inflammatory monocytes (DAPI$^{-ve}$CD45$^{+ve}$CD11b$^{+ve}$NK1.1$^{-ve}$MHCII$^{-ve}$Ly6C$^{high}$CX$_3$CR1$^{+ve}$) were sorted on an Influx cell sorter (BD Bioscience). Inflammatory and patrolling monocytes were plated in 0.1 ml of complete RPMI-medium and stimulated as indicated in the figure legends.

For RNA sequencing experiments inflammatory monocytes (DAPI$^{-ve}$CD45$^{+ve}$CD11b$^{+ve}$NK1.1$^{-ve}$Ly6C$^{high}$CX$_3$CR1$^{+ve}$CD115$^{+ve}$), patrolling monocytes (DAPI$^{-ve}$CD45$^{+ve}$CD11b$^{+ve}$NK1.1$^{-ve}$Ly6C$^{-ve}$CX$_3$CR1$^{+ve}$CD115$^{+ve}$) and neutrophils (DAPI$^{-ve}$CD45$^{+ve}$CD11b$^{+ve}$NK1.1$^{-ve}$Ly6C$^{intermediate}$Gr-1$^{high}$) were sorted from spleens of WT and ABIN1[D485N] mice.

### RNA sequencing

Total RNA was purified using RNAeasy Micro Kit (QIAGEN) according to the manufacturer's instructions and samples analysed on the Agilent Bioanalyser (#G2939AA; Agilent Technologies) with the RNA 6000 Pico Kit (#5067-1513) to assess the quality and integrity of the RNA. Libraries were prepared from each RNA sample using the NEBNext ultra RNA library prep kit for Illumina Inc. (#E7530) with the

NEBNext Poly(A) mRNA magnetic isolation module (#E7490) according to the protocol provided.

Total-RNA (40 ng) was processed with the Poly(A) mRNA magnetic isolation module to capture the mRNA fragments using oligo d(T) beads which bind to the poly(A) tail of eukaryotic mRNA. After fragmentation and priming with random hexamers, RNA templates were removed and a replacement strand synthesised incorporating dUTP in place of dTTP to generate ds cDNA. AMPure XP beads (#A63881; Beckman Coulter) were then used to separate the blunt-ended ds cDNA from the second-strand reaction mix. A single "A" nucleotide was added to the 3′ end of the blunt fragments to prevent them from ligating to one another during the subsequent adapter ligation reaction. A single "T" nucleotide at the 3′ end of the adapter provided a complementary overhang for ligation of the adapter to the fragment. Multiple indexing adapters were then ligated to the ends of the ds cDNA to prepare them for hybridisation onto a flow cell. 15 cycles of PCR were then performed to selectively enrich those DNA fragments that had adapter molecules on both ends, and to amplify the DNA in the library to the amount required for sequencing. Libraries were quantified by Qubit using the dsDNA HS assay, assessed for quality with a DNA HS Kit (#5067-4626; Agilent Technologies) and sequenced on the NextSeq 550 platform (#SY-415-1002; Illumina Inc.).

24 samples were divided between two flow cells, with biological conditions randomized between sequencing runs. Sequencing resulted in paired-end reads 2 × 75 bp, between 24 and 47 (median of 31) million reads per sample. The RNA-seq data have been deposited in ArrayExpress (www.ebi.ac.uk/arrayexpress) under accession number E-MTAB-8185. RNA-seq reads were mapped to the reference mouse genome (masked primary assembly GRCm38, release 93) using STAR 2.6.1a (Dobin et al, 2013). Typically, 80% of reads were mapped uniquely to the genome. Read counts per gene were found in the same STAR run, using Ensembl annotations in a Gene Transfer Format (GTF) file. Differential expression was performed with edgeR 3.26.4 (Robinson et al, 2010; McCarthy et al, 2012) with three contrasts (three different cell types), for each of the two genotypes (ABIN1[D485N] versus WT). A Benjamini–Hochberg multiple-test correction was applied to test *P*-values.

### Tissue processing for histological evaluation and scoring

Mice were euthanized with $CO_2$. The kidneys, liver, and lungs were removed and fixed for 48–72 h in 10% neutral buffered formalin. One kidney was trimmed longitudinally and the contralateral kidney was trimmed transversally in the mid portion. For the liver, one sample from the left lobe and two samples from the median lobe were trimmed. The lungs were processed whole without trimming. Tissues were processed to paraffin blocks, and 4-$\mu$m sections were stained with haemotoxylin and eosin (H&E) or Periodic Acid Schiff (kidney only). Kidney, liver, and lung tissue sections were assessed by a veterinary pathologist (F.M.) blinded to the genotype of the mice in the different cohorts. Ordinal semi-quantitative scoring criteria used for the assessment of lesions in the kidneys, liver, and lungs were defined according to guidelines and principles of histopathologic scoring in research (Gibson-Corley et al, 2013) and are detailed below. The photomicrographs presented in the article were captured using an Olympus BX53 with

microscope UPlanFL N objective. The ocular lens was WHN10×-H/22 and the camera used was an Olympus SC100. For preparation of the figures, images were obtained and processed using an Olympus CellSens Standard and resized using Adobe Photoshop.

### Kidney histology

Images of glomeruli were taken with objective 20×/0.50. Glomerulonephritis was defined by a variable combination of changes, including increased glomerular cellularity, size and lobulation, mesangial thickening, inflammatory cells in the glomerular tuft/mesangium, glomerulosclerosis, possible crescent formation, and possible presence of karyorrhectic debris. Glomerulonephritis was assessed in 20 glomeruli at 40-fold magnification (objective 40×/0.75 ocular 10×-H/22) according to a 0–4 ordinal grading scale: 0 = absent; 1 = mild; 2 = moderate; 3 = marked; 4 = severe.

### Liver histology

Images of liver sections were taken with objective 10×/0.30. The presence of inflammatory cell infiltration with periportal, perivascular, and/or parenchymal distribution was assessed in 20 microscopic fields at 20-fold magnification (objective 20×/0.50 ocular 10×-H/22) with the following 0–4 ordinal grading scale: 0 = absent; 1 = mild; 2 = moderate; 3 = marked; 4 = severe. Oval cell and/or bile duct proliferation was also assessed in 20 microscopic fields at 20-fold magnification with a similar 0–4 grading system. For each mouse, the average hepatic inflammation and oval cell/bile duct proliferation scores were calculated. In addition, a cumulative liver pathology score (defined as the sum of the average hepatic inflammation score and oval cell/bile duct proliferation score) was calculated for each mouse and is presented in the figures.

### Lung histology

Images of lung sections were taken with objective 10×/0.30. Lesions in the lungs, including perivascular/peribronchial inflammatory infiltrates and presence of inflammatory cells in alveolar septa were assessed in five microscopic fields at 20-fold magnification (objective 20×/0.50 ocular 10×-H/22) with the following 0–4 ordinal grading scale: 0 = absent; 1 = mild; 2 = moderate; 3 = marked; 4 = severe. An average score was calculated for each endpoint, and a cumulative lung inflammation score was calculated as the sum of the average scores of the two endpoints and is presented in the figures.

### Bone marrow chimaera experiment

WT mice expressing CD45.1 (~120 d old) were lethally irradiated (Nanda et al, 2018) and reconstituted with bone marrow cells (2.5 × 10^6 in PBS) via tail vein injection. Following reconstitution, mice received prophylactic antibiotics in their drinking water. (Levofloxacin [Sigma-Aldrich] 0.67 mg/ml for 14 d). Two groups of recipient mice were reconstituted with either CD45.2 WT cells or CD45.1 WT and CD45.2 ABIN1[D485N] cells, mixed in a 1:1 molar ratio. The animals were culled 4 mo after irradiation. The spleen was removed for flow cytometry. Kidneys and liver were collected in

fixative and processed for histological analysis. Serum was analysed for auto-antibodies.

### The IRAK4 inhibitor PF 06426779

The synthesis of PF 06426779 has been described (called compound 20 in Lee et al (2017)). PF 06426779 inhibited IRAK4 in cell-based assays with an IC50 value of 20 nM and inhibited R848-stimulated TNF secretion in human and mouse blood with IC50 values of 81 and 158 nM, respectively. The compound inhibited IRAK4 several hundred fold more potently than liver kinase B1, serine threonine kinase16, p53-related protein kinase, nuclear Dbf2-related kinase 1, and microtubule-associated serine threonine kinase 3, and >1,000-fold more potently than more than 200 other protein kinases tested, including the closely related IRAK1.

### In vivo administration of the IRAK4 inhibitor

ABIN1[D485N] and WT mice (6–8 wk old) were fed with R&M3 diet (Research Diets Inc.). Where specified, PF 06426779 was included in the diet at 4 g/kg food. Blood was collected from the tail vein at commencement of the study and 5 wk later. Mice were culled 10 wk later and blood and other tissues were analysed.

### Other methods

Cytokine measurements were carried out using Luminex-based Bio-Plex Mouse Grp 1 Cytokine 23 plex (Bio-Rad Laboratories) or individual Bio-Plex cytokine kits following the manufacturer's instructions. Autoantibodies to dsDNA and ANA (total Ig; Alpha Diagnostics International) were measured by ELISA.

### Statistical analysis

Data were analysed using GraphPad Prism 7.05 software. Data in percentages were transformed in arcsine square root before statistical analysis. The distribution was determined using the Shapiro–Wilk normality test. Pair-wise comparison of parametric and nonparametric data was done using the unpaired $t$ test with Welch's correction or unpaired Mann–Whitney test, respectively. Multiple comparisons of data with normal distribution were performed using one-way ANOVA followed by Tukey's post hoc test. Multiple comparison of nonparametric data was done using the Kruskal–Wallis test, followed by the Mann–Whitney test.

## Supplementary Information

## Acknowledgements

We thank the genotyping team of the Medical Research Council (MRC) Protein Phosphorylation and Ubiquitylation Unit for genotyping the mouse lines; Sarah Thompson for carrying out tail vein injections; and Lynn Oxford,

Lynn Stevenson, and Frazer Bell (Veterinary Diagnostic Services, School of Veterinary Medicine, University of Glasgow) for technical support with tissue processing for histological analysis. We acknowledge the expert help of Lee Murphy, Angie Fawkes, and Audrey Coutts (Wellcome Trust Clinical Research Facility, Edinburgh, UK) for generating the RNA sequencing data. The research conducted by SK Nanda, T Petrova, F Marchesi, and M Gierlinski was funded by grants from the UK Medical Research Council (MR/R021406/1) (to P Cohen and JSC Arthur) and the Wellcome Trust (209380/Z/17/Z) (to P Cohen). Some of the studies were carried out while T Petrova was the recipient of a Wellcome Trust Prize Studentship (105309/Z/14/Z). M Razsolkov holds a UK Medical Research Council studentship (MR/N013735/1). The work performed by VR Rao, K Lee, and SW Wright was funded by the Research & Development budget of Pfizer Inc.

## Author Contributions

SK Nanda: conceptualization, formal analysis, validation, investigation, visualization, methodology, and writing—original draft, review, and editing.

T Petrova: conceptualization, formal analysis, validation, investigation, visualization, methodology, and writing—original draft, review, and editing.

F Marchesi: formal analysis, investigation, visualization, methodology, and writing—review and editing.

M Gierlinski: data curation, software, visualization, methodology, and writing—review and editing.

M Razsolkov: formal analysis and investigation.

KL Lee: resources.

SW Wright: resources.

VR Rao: resources, funding acquisition, and writing—review and editing.

P Cohen: conceptualization, supervision, funding acquisition, project administration, and writing—original draft, review, and editing.

JSC Arthur: conceptualization, supervision, funding acquisition, project administration, and writing—review and editing.

## Conflict of Interest Statement

The authors declare that they have no conflict of interest.

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
