## [Reviewer comments · Life Science Alliance]

Life Science Alliance

Distinct signals and immune cells drive liver pathology and glomerulonephritis in ABIN1[D485N] mice

Sambit Nanda, Tsvetana Petrova, Francesco Marchesi, Marek Gierlinski, Momchil Razsolkov, Katherine Lee, Stephen Wright, Vikram Rao, Philip Cohen, and Simon Arthur

DOI: <https://doi.org/10.26508/lsa.201900533>

Corresponding author(s): Philip Cohen, Dundee, University of and Simon Arthur, University of Dundee

Review Timeline:	Submission Date:	2019-08-23
	Editorial Decision:	2019-09-17
	Revision Received:	2019-10-16
	Editorial Decision:	2019-10-18
	Revision Received:	2019-10-24
	Accepted:	2019-10-24

Scientific Editor: Andrea Leibfried

Transaction Report:

September 17, 2019

Re: Life Science Alliance manuscript #LSA-2019-00533-T

Prof. Philip Cohen
Dundee, University of
MRC Protein Phosphorylation and Ubiquitylation Unit
School of Life Sciences
University of Dundee
Dundee, Scotland DD1 5EH
United Kingdom

Dear Dr. Cohen,

Thank you for submitting your manuscript entitled "The myddosome drives all the facets of lupus in mice expressing a ubiquitin-binding-defective mutant of ABIN1" to Life Science Alliance. The manuscript was assessed by expert reviewers, whose comments are appended to this letter.

As you will see, the reviewers appreciate your analyses and provide constructive input on how to further strengthen your study. We would thus like to invite you to submit a revised version to us, following the suggestions made by the reviewers. Doing so seems rather straightforward, but please get in touch in case you would like to discuss an individual revision point further.

Thank you for this interesting contribution to Life Science Alliance. We are looking forward to receiving your revised manuscript.

Sincerely,

Andrea Leibfried, PhD

B. MANUSCRIPT ORGANIZATION AND FORMATTING:

Reviewer #1 (Comments to the Authors (Required)):

In the current manuscript, Nanda et al. demonstrate that MyD88-IRAK4/IRAK1 signaling is critical for the development of lupus in ABIN1(D485NN) mutant mice. They show that IL-6 is important for the development of glomerulonephritis but not for liver or lung pathology. Also, in ABIN1(D485NN) x TRL7 KO mice, glomerulonephritis was almost prevented while liver and lung pathology were only reduced but not eliminated. Equally, they showed that lymphocytes are required for glomerulonephritis (using ABIN1(D485NN) x RAG2 KO mice) but liver pathology is independent of the adaptive immune system. Interestingly, they find a drastic increase (independent of IL-6 and lymphocytes) in patrolling monocytes in the blood and also in the spleen, liver, kidney and lungs.

It is a very well written manuscript with clear study goals and solid experimental design. The authors demonstrate that different components of the immune system are responsible for some of the clinical manifestations of the disease model and interestingly suggest that different organs rely on different mechanisms/cell types to drive pathology.

The manuscript somewhat lacks novelty, as the authors have previously demonstrated that both MyD88 and IRAK(s) are involved in the development of the disease and another previous publication suggested that monocytes play a role in the pathogenesis of lupus (Kuriakose 2019). Thus, it is important to better distinguish their paper from previously published reports. Perhaps they can better emphasize the differences in liver vs. kidney pathology they have found and speculate about mechanisms?

The finding that monocyte frequencies in the blood and numbers in the spleen increase dramatically before the onset of disease is interesting suggesting that they are the drivers of the disease. Yet, this warrants further investigation in terms of mechanistic involvement in disease development, as well as a potential prognostic biomarker. Have they for example tried to deplete monocytes in their model or crossed their mice to monocyte-deficient mice? Would this have an impact on liver pathology?

The title should be changed to more accurately reflect the findings of the study. The myddosome as key driver of this disease model was already shown by the authors in their study in 2011. I understand that this manuscript shows more detailed characterization of the lung and liver pathology compared to their previous study, but nevertheless the title does not reflect the major findings of the present study.

Fig. 5: ABIN1(D485N) mice appear to have severely reduced frequencies of NK-cells and neutrophils in all organs investigated. Although this may be due to the relative increase in monocytes, the authors nevertheless need to include frequencies and numbers of these cells (and other immune cells, e.g. monocyte-derived cells, eosinophils, macrophages, lymphocytes etc. in the different organs), and comment on how this may relate to their phenotype.

Minor comments:

The authors enumerate cells as $\times 10^{-6}$ rather than $\times 10^6$

Fig. 4B, the relative frequencies of Ly6Chi to Ly6Clo monocytes in the blood of WT mice in their study is highly skewed towards Ly6Clo monocytes according to the flow plot. However the cells are usually at roughly equal frequency among blood monocytes (Hanna, NI, 2011) and this is reflected in the summary histograms in 4D and F. The authors should replace the flow plot with a more representative of the data.

Fig. 5, flow plots are missing WT and Mutant labels.

Line 239: '...larger larger...'

Line 318: For clarity, the sentence should say anti-IL6 antibodies or antibodies targeting the IL-6 pathway, rather than IL-6 antibodies.

Lin 154: From their results with the RAG2 KO mice and BM chimeras they conclude that a critical

role for myeloid cells can be shown. Yet, from these experiments, they can only conclude that it is a hematopoietic and non-T or non-B cell. See above, what are the total cell numbers of NK cells, neutrophils, lymphocytes, etc. in the different organs?

Reviewer #2 (Comments to the Authors (Required)):

Review 31st August 2019

- You should be trying to help the work get published not necessarily in this journal but ultimately.
 - Don't criticize an experiment unless you can tell the authors how they could do it better. "If you just want to throw darts," he would say, "go to the pub."
 - Keep in mind that no one ever built a statue to a critic.
 - Try to act as a peer in the process of peer review.
- Science Signaling 2009 Michael Yaffe

Title: The myddosome drives all the facets of lupus in mice expressing a ubiquitin-binding-defective mutant of ABIN1

Manuscript # LSA-2019-00533-T

General Remarks

This excellent study shows that the phenotype of ABIN1[D485N] that are prone to a Lupus like syndrome is driven by overproduction of IL6 in part and by IRAK4-Myd88 signalling specifically. The genetic analysis is very clear cut and the differences very obvious. They also analyse or examine the role of the innate and adaptive immune system in the phenotype. Finally they show that pharmacological inhibition of IRAK4 strongly ameliorates the lupus like phenotype in the ABIN1[D485N] mice giving a nice touch of clinical relevance. I found the manuscript very easy to read and follow and the authors can be congratulated on a very good job.

I have the most minor comments

line 248 should be "led us to study the" no OF

line 239 should be "larger" only once

Fig. 6e-I it appears to me that the second set of "significance" bars in these panels have slipped, as far as I can see the relevant comparison (IL6 ko {plus minus} D485N) is not the one that is indicated.

Reviewer#1

It is a very well written manuscript with clear study goals and solid experimental design. The authors demonstrate that different components of the immune system are responsible for some of the clinical manifestations of the disease model and interestingly suggest that different organs rely on different mechanisms/cell types to drive pathology.

Response: We thank the reviewer for the positive comments made about our paper.

1. The manuscript somewhat lacks novelty, as the authors have previously demonstrated that both MyD88 and IRAK(s) are involved in the development of the disease and another previous publication suggested that monocytes play a role in the pathogenesis of lupus (Kuriakose 2019). Thus, it is important to better distinguish their paper from previously published reports. Perhaps they can better emphasize the differences in liver vs. kidney pathology they have found and speculate about mechanisms?

Response: In the revised manuscript, we now include in the Results section new information about the different types of immune cells present in the liver and kidneys of ABIN1[D485N] and WT mice (presented in the new Figs S5 and S6), and have rewritten and expanded a section of the Discussion which speculates about the mechanisms that may drive the liver pathology, based in part on the new results we have obtained during revision.

2. The finding that monocyte frequencies in the blood and numbers in the spleen increase dramatically before the onset of disease is interesting suggesting that they are the drivers of the disease. Yet, this warrants further investigation in terms of mechanistic involvement in disease development, as well as a potential prognostic biomarker. Have they for example tried to deplete monocytes in their model or crossed their mice to monocyte-deficient mice? Would this have an impact on liver pathology?

Response: We have injected purified patrolling monocytes from 16 week-old ABIN1[D485N] mice into WT mice to investigate if they would drive aspects of the lupus pathology, but this proved to be impossible because the monocytes did not survive in the WT mice for more than a week. We have also depleted monocytes and macrophages by treating mice with Clodronate, but the effect is transient and the cells return to normal levels after 5-7 days. To conduct an experiment similar to that shown in Fig 8, would require Clodronate to be injected into ABIN1[D485N] mice every few days for 10 weeks to see if the liver pathology was prevented. We have not carried out this experiment because we also found that even one treatment of WT mice with Clodronate increases the number of neutrophils in the blood, spleen and lungs, which only return to lower levels after a week or longer. Therefore

experiments involving repeated Clodronate injections would compromise the experiment and complicate interpretation of the data as neutrophil NETosis may contribute to the pathology.

[A paragraph of the author response to the reviewer describing future plans has been removed by LSA staff at the request of the authors and in accordance with LSA policies.]

As the reviewer mentions, genetic depletion of the monocytes would require crossing the ABIN1[D485N] mice to Nr4a1 KO mice. We are not confident that this experiment will be definitive, because recent papers have revealed that in Nr4a1 KO mice or after siRNA knockdown of the mRNA encoding this protein, monocyte-derived DCs display enhanced inflammatory responses, and T cell activation and proliferation are also enhanced for this and other reasons (Tel-Karthauss et al, 2018, Front. Immunol. 9, 1797; Liebmann et al, 2018, PNAS 115, E8017-8026; Liu et al, Nature 567, 2019, 525-529). There is also marked expansion of IgM Plasma Cells of B-1a origin (Huizar et al, 2017, Immuno-horizons 1, 188-197). The liver pathology may therefore be altered in unpredictable ways. Moreover, this experiment would take at least 2 years to complete (import of the mice into our facility, crossing to the ABIN1[D485N] mice, colony expansion, and aging of the mice for six months, followed by analysis of the liver pathology and other organ pathologies).

3. The title should be changed to more accurately reflect the findings of the study. The myddosome as key driver of this disease model was already shown by the authors in their study in 2011. I understand that this manuscript shows more detailed characterization of the lung and liver pathology compared to their previous study, but nevertheless the title does not reflect the major findings of the present study.

Response. We have changed the title to more accurately reflect the findings of our study and we have also rewritten the summary to reflect the increased emphasis of the paper on the different pathways driving the pathology of the liver and kidney.

4. Fig. 5: ABIN1(D485N) mice appear to have severely reduced frequencies of NK-cells and neutrophils in all organs investigated. Although this may be due to the relative increase in monocytes, the authors nevertheless need to include frequencies and numbers of these cells (and other immune cells, e.g. monocyte-derived cells, eosinophils, macrophages, lymphocytes etc. in the different organs), and comment on how this may relate to their phenotype.

Response: We thank the reviewer for this excellent suggestion. We now include the frequencies and numbers of the NK cells, neutrophils, eosinophils, monocyte-derived dendritic cells, macrophages, T cells and their state of activation and B cells in the different organs. The results are presented in the new Figs S5 and S6 and in an additional section of the Results. These experiments have revealed some interesting differences between the three organs of

the ABIN1[D485N] mice studied, some of which are included in a rewritten and expanded section of the Discussion.

5. Minor comments:

(a) The authors enumerate cells as $\times 10^{-6}$ rather than $\times 10^6$ (?).

Response-We think that the ordinates, showing the number of cells, are labelled correctly. For example, in Figure 1B, the GCB cell numbers are plotted as 0.5, 1.0, 1.5, 2.0 and 2.5 because this is one millionth (10^{-6}) of the number of GCB cells, as is indicated on the ordinate.

(b) Fig. 4B, the relative frequencies of Ly6Chi to Ly6Clo monocytes in the blood of WT mice in their study is highly skewed towards Ly6Clo monocytes according to the flow plot. However the cells are usually at roughly equal frequency among blood monocytes (Hanna, NI, 2011) and this is reflected in the summary histograms in 4D and F. The authors should replace the flow plot with a more representative of the data.

Response: As requested by the reviewer, we have replaced the Flow plots in Figure 4B with more representative data.

(c) Fig. 5, flow plots are missing WT and Mutant labels

Response: we have added the missing WT and Mutant labels).

Line 239: '...larger larger...'

Response: We have corrected this error

Line 318: For clarity, the sentence should say anti-IL6 antibodies or antibodies targeting the IL-6 pathway, rather than IL-6 antibodies

Response: we agree and have corrected the wording

Line 154: From their results with the RAG2 KO mice and BM chimeras they conclude that a critical role for myeloid cells can be shown. Yet, from these experiments, they can only conclude that it is a hematopoietic and non-T or non-B cell. See above, what are the total cell numbers of NK cells, neutrophils, lymphocytes, etc. in the different organs?

Response: we agree and have changed "myeloid cells" to non-T, non-B cells at this point in the text. The total numbers of each immune cell in the different organs are presented in the new Figs S5 and S6 (see response above to the the 4th major point raised by the reviewer.

Reviewer #2

Manuscript # LSA-2019-00533-T

General Remarks

This excellent study shows that the phenotype of ABIN1[D485N] that are prone to a Lupus like syndrome is driven by overproduction of IL6 in part and by IRAK4-Myd88 signalling specifically. The genetic analysis is very clear cut and the differences very obvious. They also analyse or examine the role of the innate and adaptive immune system in the phenotype. Finally they show that pharmacological inhibition of IRAK4 strongly ameliorates the lupus like phenotype in the ABIN1[D485N] mice giving a nice touch of clinical relevance. I found the manuscript very easy to read and follow and the authors can be congratulated on a very good job.

Response:- We thank the reviewer for the kind comments.

I have the most minor comments
line 248 should be "led us to study the" no OF

Response:- The error noticed by Reviewer 2 has been corrected.

line 239 should be "larger" only once

Response:- The error, also noticed by Reviewer 1, has been corrected.

Fig. 6e-l it appears to me that the second set of "significance" bars in these panels have slipped, as far as I can see the relevant comparison (IL6 ko {plus minus} D485N) is not the one that is indicated.

Response:- The second set of significance bars are comparing the significance of the data sets in lanes 2 and 3 and are presented correctly. We have made minor modifications to the Figure Legend to clarify this point and to mention that no horizontal bars are shown for results that are not statistically significant.

October 18, 2019

RE: Life Science Alliance Manuscript #LSA-2019-00533-TR

Prof. Philip Cohen
Dundee, University of
MRC Protein Phosphorylation and Ubiquitylation Unit
School of Life Sciences
University of Dundee
Dundee, Scotland DD1 5EH
United Kingdom

Dear Dr. Cohen,

Thank you for submitting your revised manuscript entitled "Distinct signals and immune cells drive liver pathology and glomerulonephritis in ABIN1[D485N] mice". I have now re-assessed your manuscript and point-by-point response to the concerns raised by the reviewers. I think your responses address the concerns well and I'd be thus happy to publish your paper in Life Science Alliance, pending final revisions necessary to meet our formatting guidelines:

- please add callouts in the manuscript text to Fig. 4A, Fig. S7D, Fig. S8C-H
- please mention Fig S8G-H in the legend for this figure

A. FINAL FILES:

-- Summary blurb (enter in submission system): A short text summarizing in a single sentence the study (max. 200 characters including spaces). This text is used in conjunction with the titles of

papers, hence should be informative and complementary to the title. It should describe the context and significance of the findings for a general readership; it should be written in the present tense and refer to the work in the third person. Author names should not be mentioned.

B. MANUSCRIPT ORGANIZATION AND FORMATTING:

Sincerely,

October 24, 2019

RE: Life Science Alliance Manuscript #LSA-2019-00533-TRR

Prof. Philip Cohen
Dundee, University of
MRC Protein Phosphorylation and Ubiquitylation Unit
School of Life Sciences
University of Dundee
Dundee, Scotland DD1 5EH
United Kingdom

Dear Dr. Cohen,

Thank you for submitting your Research Article entitled "Distinct signals and immune cells drive liver pathology and glomerulonephritis in ABIN1[D485N] mice". It is a pleasure to let you know that your manuscript is now accepted for publication in Life Science Alliance. Congratulations on this interesting work.

*****IMPORTANT:** If you will be unreachable at any time, please provide us with the email address of an alternate author. Failure to respond to routine queries may lead to unavoidable delays in publication.*******

DISTRIBUTION OF MATERIALS:

Again, congratulations on a very nice paper. I hope you found the review process to be constructive and are pleased with how the manuscript was handled editorially. We look forward to future exciting

submissions from your lab.

Sincerely,
